# Study of 300,486 individuals identifies 148 independent genetic loci influencing general cognitive function

Gail Davies[1], Max Lam et al.[#]

General cognitive function is a prominent and relatively stable human trait that is associated with many important life outcomes. We combine cognitive and genetic data from the CHARGE and COGENT consortia, and UK Biobank (total $N = 300,486$; age 16–102) and find 148 genome-wide significant independent loci ($P < 5 \times 10^{-8}$) associated with general cognitive function. Within the novel genetic loci are variants associated with neurodegenerative and neurodevelopmental disorders, physical and psychiatric illnesses, and brain structure. Gene-based analyses find 709 genes associated with general cognitive function. Expression levels across the cortex are associated with general cognitive function. Using polygenic scores, up to 4.3% of variance in general cognitive function is predicted in independent samples. We detect significant genetic overlap between general cognitive function, reaction time, and many health variables including eyesight, hypertension, and longevity. In conclusion we identify novel genetic loci and pathways contributing to the heritability of general cognitive function.

[#]A full list of authors and their affliations appears at the end of the paper.

Some individuals have generally higher cognitive function than others. These individual differences are quite persistent across the life course from later childhood onwards. Individuals with higher measured general cognitive function tend to live longer and be less deprived. Retaining general cognitive function is an important aspect of healthy ageing. The population variance in this medically- and socially-important trait has environmental and genetic aetiologies. The details of the genetic contributions are, as-yet, poorly understood.

Since the discovery of general cognitive ability (or 'g') in 1904[1], hundreds of studies have replicated the finding that around 40% of the variance in subjects' scores on a diverse battery of cognitive tests can be accounted for by a single general factor[2]. Some variance is also attributable to individual cognitive domains (e.g., reasoning, memory, processing speed, and spatial ability), and some is attributable to specific cognitive skills associated with individual mental tests. However, all cognitive tests rely to a greater or lesser extent on general cognitive ability for successful execution. Figure 1 illustrates and explains this hierarchical model of cognitive ability differences[3]. Therefore, using a general cognitive function phenotype in a genetically-informative design is supported by the observation that the well-established positive manifold of cognitive tests may be represented by a substantially heritable, higher-order, latent general cognitive function phenotype[2,4,5].

There are two commonly-used routes that are used to obtain general cognitive ability scores for each participant in a sample. First, if all members of a sample have taken the same set of diverse cognitive tests, then a data reduction procedure (such as principal components analysis (PCA) or factor analysis) can be applied. Typically, this finds that all tests load on (i.e., correlate positively with) the first unrotated component, or factor, and scores on this component can be calculated for each person; this gives each person a g score. Second, some mental tests—usually those involving complex mental work, and often those with a variety of item types—have a high g loading[2]. That is, scores on some individual cognitive tests can be used to obtain an acceptable proxy for general cognitive ability. An example of the latter is the Moray House Test of verbal and numerical reasoning, which has a high correlation with a PCA-derived general cognitive function score[6].

General cognitive function is peerless among human psychological traits in terms of its empirical support and importance for life outcomes[7,8]. Individuals who have higher cognitive function in childhood and adolescence tend to stay longer in education, gain higher educational qualifications, progress to more professional and better-paid jobs, live healthier lives, and live longer. Individual differences in general cognitive function show phenotypic and genetic stability across most of the life course[9–11]. The phenotypic correlation between general cognitive function scores on the same people at age 11 and age 70–80 years is almost 0.7, and remains above 0.5 when age 11 versus age 90 scores are correlated.

Twin studies find that general cognitive function has a heritability of more than 50% from adolescence through adulthood to older age[4,5,12]. SNP-based estimates of heritability for general cognitive function are about 20–30%[13]. However, these estimates might increase to about 50% when family-based designs are used to retain the contributions made by rarer SNPs[14]. To date, little of this substantial heritability has been explained, i.e., only a few relevant genetic loci have been discovered (Table 1; Supplementary Fig. 1). As has been found with other highly polygenic traits, a limitation on uncovering relevant genetic loci is sample size[15]; to date, there have been fewer than 100,000 individuals in studies of general cognitive function[13,16]. The MTAG (multi-trait analysis of genome-wide association studies) method has been used to corral cognitive function and associated traits to expand the number of loci associated with general cognitive function[17]. However, the present study uses only cognitive function phenotypes, and amasses a total sample size of over 300,000.

The present study also tests for genetic contributions to reaction time, and examines its genetic relationship with general cognitive function. Reaction time is both phenotypically and genetically correlated with general cognitive function, and accounts for some of its association with health[18–20]. By making these comparisons between general cognitive function and reaction time, we identify regions of the genome that have a shared correlation with general cognitive function and more elementary cognitive tasks[21].

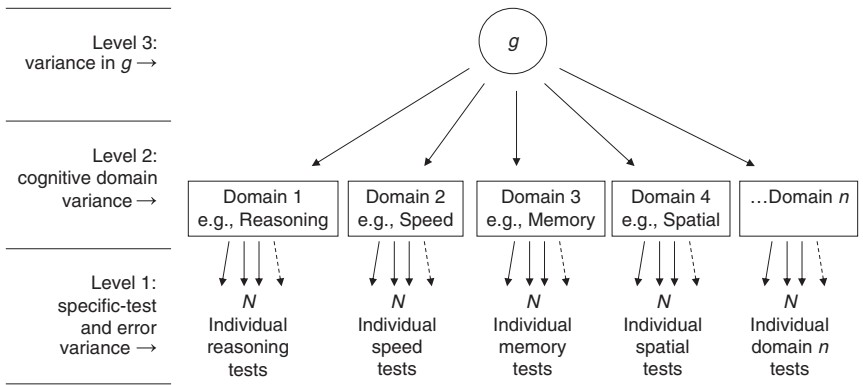

**Fig. 1** The hierarchical model of cognitive function variance. At level 1, individuals differ in specific tests that assess the various cognitive domains. Scores on all the tests correlate positively. It is found that there are especially strong correlations among the tests of the same domain, so a latent trait at the domain level can be extracted to represent this common variance. It is then found that individuals who do well in one domain also tend to do well in the other domains, so a general cognitive latent trait called g can be extracted. This model allows researchers to partition cognitive performance variance into these different levels. They can then explore the causes and consequences of variance at different levels of cognitive specificity-generality. For example, there are genetic and ageing effects on g and on some specific domains, such as memory and speed of processing. Note that the specific-test-level variance contains variation in the performance of skills that are specific to the individual test and also contains error variance. (Reproduced, with permission, from ref. [3])

**Table 1 Details of GWA studies of general cognitive function to date, including the present study**

| Author; doi | Year | N | GWAS-sig SNP hits | GWAS-sig gene hits | SNP-based $h^2$ |
|---|---|---|---|---|---|
| Davies et al. (2011)[86] | 2011 | 3511 | 0 | 1 gene | 0.51 (0.11) |
| Lencz et al. (2013)[87] | 2013 | 5000 | 0 | NA | NA |
| Benyamin et al. (2014)[88] | 2014 | 17,989 | 0 | 0 | 0.46 (0.06) |
| Kirkpatrick et al. (2014)[89] | 2014 | 7100 | 0 | 0 | 0.35 (0.11) |
| Davies et al. (2015)[25] | 2015 | 53,949 | 3 loci (13 SNPs) | 1 gene | 0.29 (0.05) |
| Davies et al. (2016); results for 'fluid' test | 2016 | 36,035 | 3 loci (149 SNPs) | 7 loci 17 genes | 0.31 (0.02) |
| Trampush et al. (2017)[64] | 2017 | 35,298 | 2 loci (7 SNPs) | 3 loci 7 genes | 0.22 (0.01) |
| Sniekers et al. (2017)[16] | 2017 | 78,308 | 18 loci (336 SNPs) | 47 genes | 0.20 (0.01) |
| Davies et al. (2018); present study | 2018 | 300,486 | 148 loci (11,600 SNPs) | 709 genes | 0.25 (0.006) |

For SNP-based heritability, the value from the largest sample is given

## Results

**General cognitive function phenotypes**. The psychometric characteristics of the general cognitive component from each cohort in the CHARGE consortium are shown in Supplementary Note 1. In order to address the fact that different cohorts had applied different cognitive tests, we previously showed that two general cognitive function components extracted from different sets of cognitive tests on the same participants correlate highly[13]. The cognitive test from the large UK Biobank sample was the so-called 'fluid' test, a 13-item test of verbal-numerical reasoning, which has a high genetic correlation with general cognitive function[22]. With the CHARGE and COGENT samples' general cognitive function scores and UK Biobank's verbal-numerical reasoning scores, there were 300,486 participants included in the present report's meta-analysis of genome-wide association studies (GWASs). Note that we included four UK Biobank samples, i.e. three assessment centre-tested samples, and one online-tested sample. The genetic correlation between CHARGE's-COGENT's general cognitive function component and UK Biobank's verbal-numerical reasoning test, calculated for the present study using linkage disequilibrium score (LDSC) regression, was estimated at 0.87 (SE = 0.03). This indicates very substantial overlap between the genetic variants associated with cognitive function in these two groups.

**SNP-based meta-analyses of cognitive function GWASs**. We performed an N-weighted meta-analysis of general cognitive function which included all of the CHARGE, COGENT, and UK Biobank samples. Meta-analysis of the results for the general cognitive function GWASs found 11,600 significant ($P < 5 \times 10^{-8}$) SNP associations, and 21,855 at a suggestive level ($1 \times 10^{-5} > P \geq 5 \times 10^{-8}$); see Fig. 2a, Supplementary Fig. 2a, and Supplementary Data 1 and 2. There were 434 'independent' significant SNPs; see Methods section for description of independent SNP selection criteria, distributed within 148 loci across all autosomal chromosomes. Note that, for consistency, we use the term 'independent' here according to the definition that is used in the relevant analysis package. A comparison of these 148 loci with results from the largest previous GWASs of cognitive function[16], and educational attainment[24], and an MTAG analysis of cognitive function[17]—all of which included a subsample of individuals contributing to the present study—confirmed that 11 of 18, 24 of 74, and 89 of 187 of these were, respectively, genome-wide significant in the present study (Supplementary Data 3). Of the 148 loci found in the present study, 58 have not been reported previously in other GWA studies of cognitive function or educational attainment (novel loci are indicated in Supplementary Data 4). One hundred and seventy-eight lead SNPs were identified within these 148 loci.

For the 434 independent significant SNPs and tagged SNPs, a summary of previous SNP associations is listed in Supplementary Data 5. They have been associated with many physical (e.g., BMI, height, weight), medical (e.g., lung cancer, Crohn's disease, blood pressure), and psychiatric (e.g., bipolar disorder, schizophrenia, autism) traits. Of the 58 new loci, we highlight previous associations with schizophrenia (2 loci), Alzheimer's disease (1 locus), and Parkinson's disease (1 locus).

We sought to identify independent significant and tagged SNPs within the 148 significant genomic risk loci associated with general cognitive function that are potentially functional (Fig. 3a; Supplementary Data 4). See Methods section for further details. Across many of the loci there is clear evidence of functionality including involvement in gene regulation, deleterious SNPs, eQTLs, and regions of open chromatin.

**General cognitive function gene-based and gene-set results**. A gene-based association analysis identified 709 genes as significantly associated with general cognitive function (Fig. 2b; Supplementary Fig. 2b; Supplementary Data 6). These 709 genes were compared to gene-based associations from previous studies of general cognitive function and educational attainment[13,16,17,25]; 418 were replicated in the present study, and 291 were novel. The 291 new gene-based associations are highlighted in Supplementary Data 6. Several of the specific genes associated with general cognitive function are considered in detail in the Discussion, below.

Gene-set analysis identified seven significant gene sets associated with general cognitive function: neurogenesis ($P = 1.57 \times 10^{-9}$), regulation of nervous system development ($P = 7.52 \times 10^{-7}$), neuron projection ($P = 7.89 \times 10^{-7}$), positive regulation of nervous system development ($P = 9.42 \times 10^{-7}$), neuron differentiation ($P = 1.68 \times 10^{-6}$), regulation of cell development ($P = 1.93 \times 10^{-6}$), and dendrite ($P = 3.52 \times 10^{-6}$) (Supplementary Data 7). Gene-property analysis can show if tissue-specific expression levels are associated with a gene's association with a phenotype. This analysis indicated a significant association between transcription levels in all brain regions—except the brain spinal cord and cervical c1—and the association with general cognitive function. In addition, expression levels in the pituitary were associated with gene-based association with general cognitive function; these results indicate that the genes with the highest expression levels in these regions were those showing the greatest associations with general cognitive function. (Fig. 3b, c; Supplementary Table 1; Supplementary Data 8). The significance of this relationship was greatest in the cerebellum and the cortex.

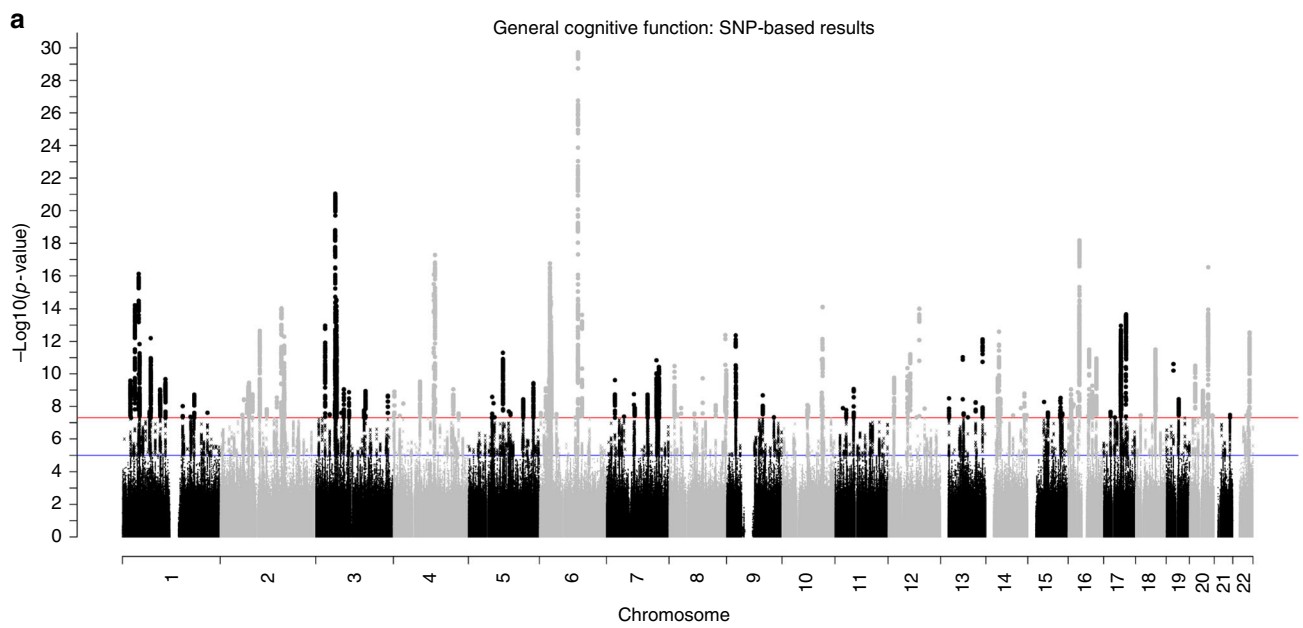

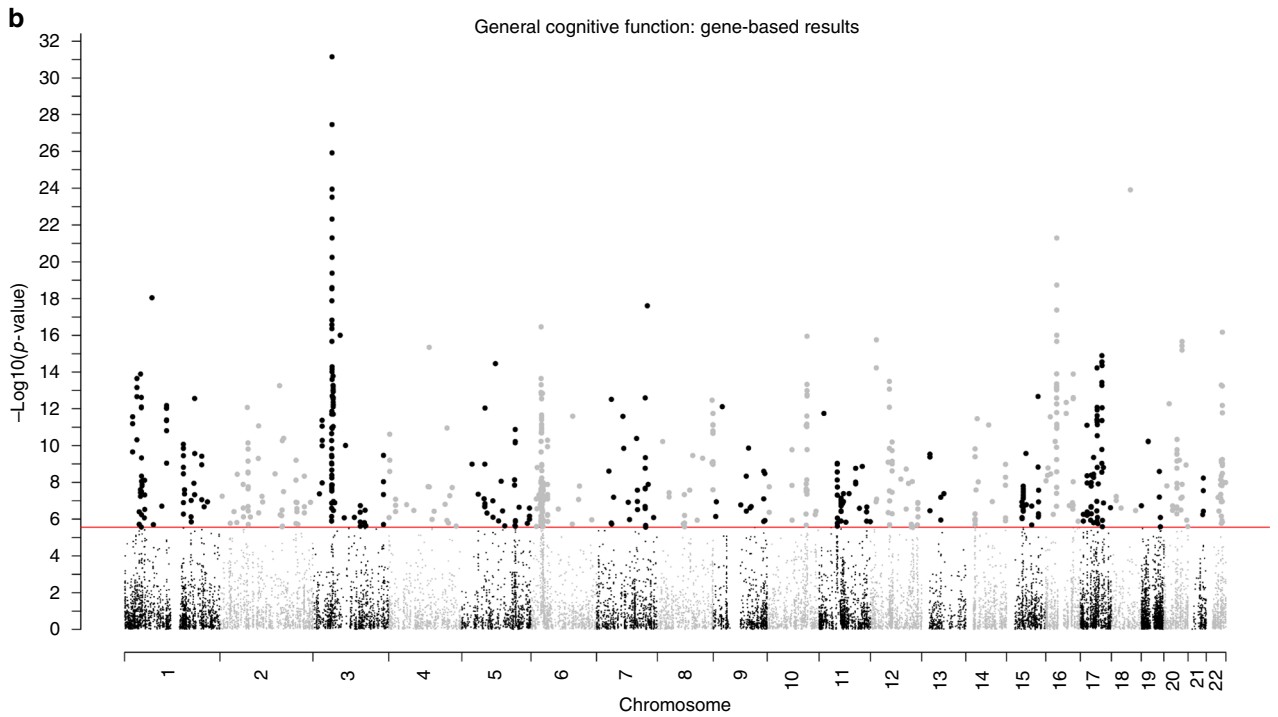

**Fig. 2** Association results for general cognitive function. SNP-based (**a**) and gene-based (**b**) association results in 300,486 individuals. The red line indicates the threshold for genome-wide significance: $P < 5 \times 10^{-8}$ for (**a**), $P < 2.75 \times 10^{-6}$ for (**b**); the blue line in (**a**) indicates the threshold for suggestive significance: $P < 1 \times 10^{-5}$

**SNP-based heritability of general cognitive function**. We estimated the proportion of variance explained by all common SNPs using GCTA-GREML in four of the largest individual samples: English Longitudinal Study of Ageing (ELSA: $N = 6661$, $h^2 = 0.12$, SE = 0.06), Understanding Society ($N = 7841$, $h^2 = 0.17$, SE = 0.04), UK Biobank Assessment Centre ($N = 86,010$, $h^2 = 0.25$, SE = 0.006), and Generation Scotland ($N = 6,507$, $h^2 = 0.20$, SE = 0.05[23]) (Table 2). Genetic correlations for general cognitive function amongst these cohorts, estimated using bivariate GCTA-GREML, ranged from $r_g = 0.88$ to 1.0 (Table 2). These results indicate that the same genetic variants contribute to

phenotypic differences in general cognitive function across each of these three samples. We investigated the genetic contribution to the stability of individual differences in people's verbal-numerical reasoning, by examining data from those individuals in UK Biobank who completed the test on two occasions (mean time gap = 4.93 years). We found a significant and perfect genetic correlation of $r_g = 1.0$ (SE = 0.02).

**Polygenic profile scores and genetic correlations**. After omitting them from the meta-analysis of GWASs, we created general cognitive function polygenic profile scores in three

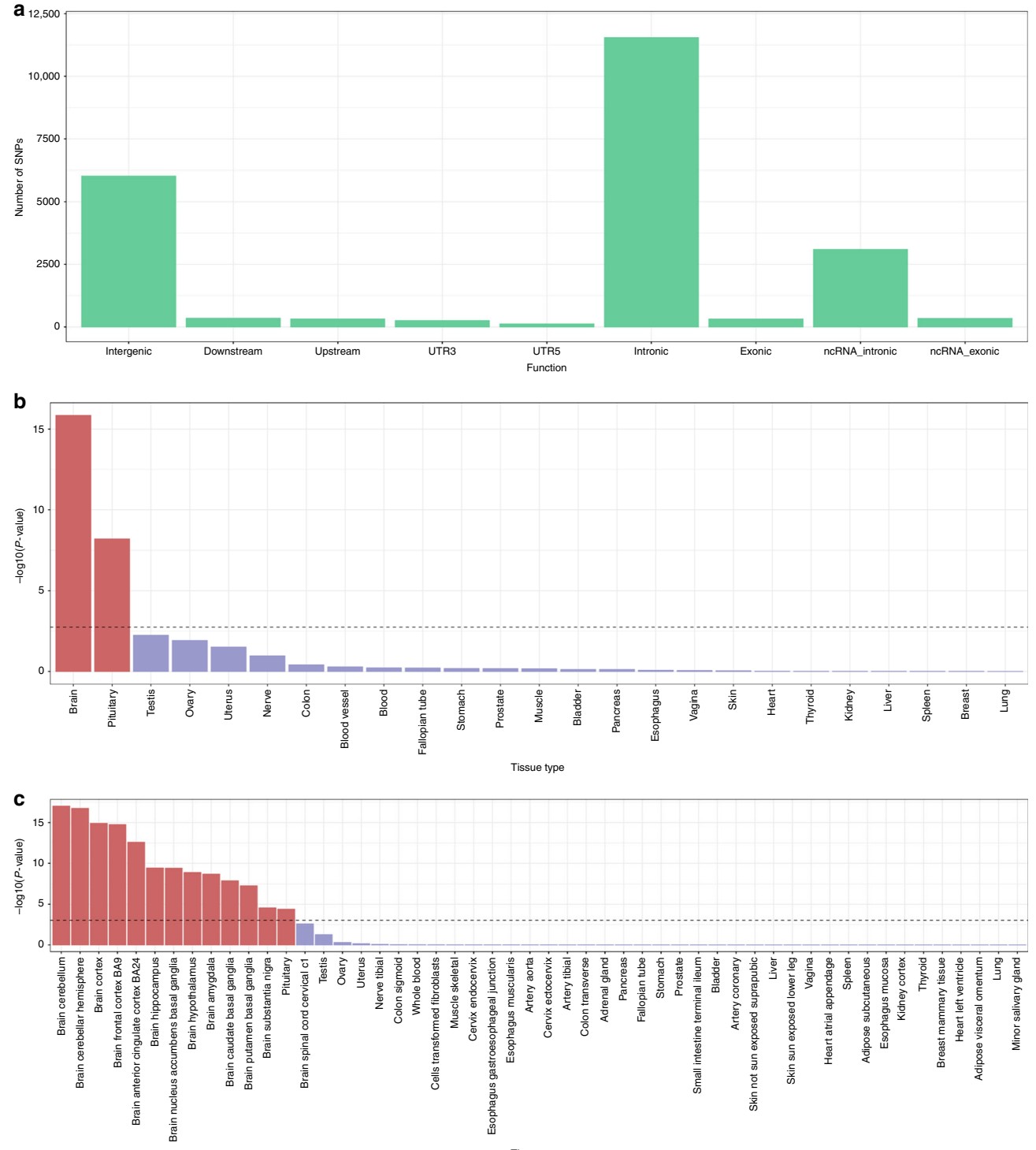

**Fig. 3** Functional analyses of general cognitive function. Analyses include general cognitive function-associated SNPs, independent significant SNPs, and all SNPs in LD with independent significant SNPs. Functional consequences of SNPs on genes (**a**) indicated by functional annotation assigned by ANNOVAR. MAGMA gene-property analysis results; results are shown for average expression of 30 general tissue types (**b**) and 53 specific tissue types (**c**). The dotted line indicates the Bonferroni-corrected α level

of the larger cohorts: ELSA, Generation Scotland, and Understanding Society. The polygenic profile score for general cognitive function explained 2.63% of the variance in ELSA ($\beta = 0.17$, SE = 0.01, $P = 1.70 \times 10^{-51}$), 3.73% in Generation Scotland ($\beta = 0.20$, SE = 0.01, $P = 5.02 \times 10^{-68}$), and 4.31% in Understanding Society ($\beta = 0.22$, SE = 0.01, $P = 6.17 \times 10^{-88}$). Full results for all five thresholds are shown in Supplementary Table 2.

We tested the genetic correlations between general cognitive function and 52 health-related traits. Thirty-six of these health-related traits were significantly genetically correlated with general cognitive function (Supplementary Data 9). We report significant genetic correlations between general cognitive function and: hypertension ($r_g = -0.15$, SE = 0.02), grip strength (right hand: $r_g = 0.09$, SE = 0.02), wearing glasses or

**Table 2 Genetic correlations and heritability estimates of a general cognitive function component in three United Kingdom cohorts**

| Cohort | ELSA | US | GS |
|---|---|---|---|
| ELSA | 0.12 (0.06) | | |
| US | 1.0 (0.33) | 0.17 (0.04) | |
| GS | 1.0 (0.38) | 0.88 (0.24) | 0.20 (0.05) |

Below the diagonal, genetic correlations (standard error) of general cognitive function amongst three cohorts are shown: *ELSA* English Longitudinal Study of Ageing, *GS* Generation Scotland, *US* Understanding Society. SNP-based heritability (standard error) estimates appear on the diagonal

contact lenses ($r_g = 0.28$, SE = 0.04), short-sightedness ($r_g = 0.32$, SE = 0.03), long-sightedness ($r_g = -0.21$, SE = 0.05), heart attack ($r_g = -0.17$, SE = 0.03), angina ($r_g = -0.18$, SE = 0.03), lung cancer ($r_g = -0.26$, SE = 0.05), and osteoarthritis ($r_g = -0.24$, SE = 0.04). We also report a significant genetic correlation with major depressive disorder ($r_g = -0.30$, SE = 0.04); this result strengthens previously-reported non-significant correlations of around $-0.10$[16,17]. We also note the important genetic association between general cognitive function and longevity ($r_g = 0.17$, SE = 0.06).

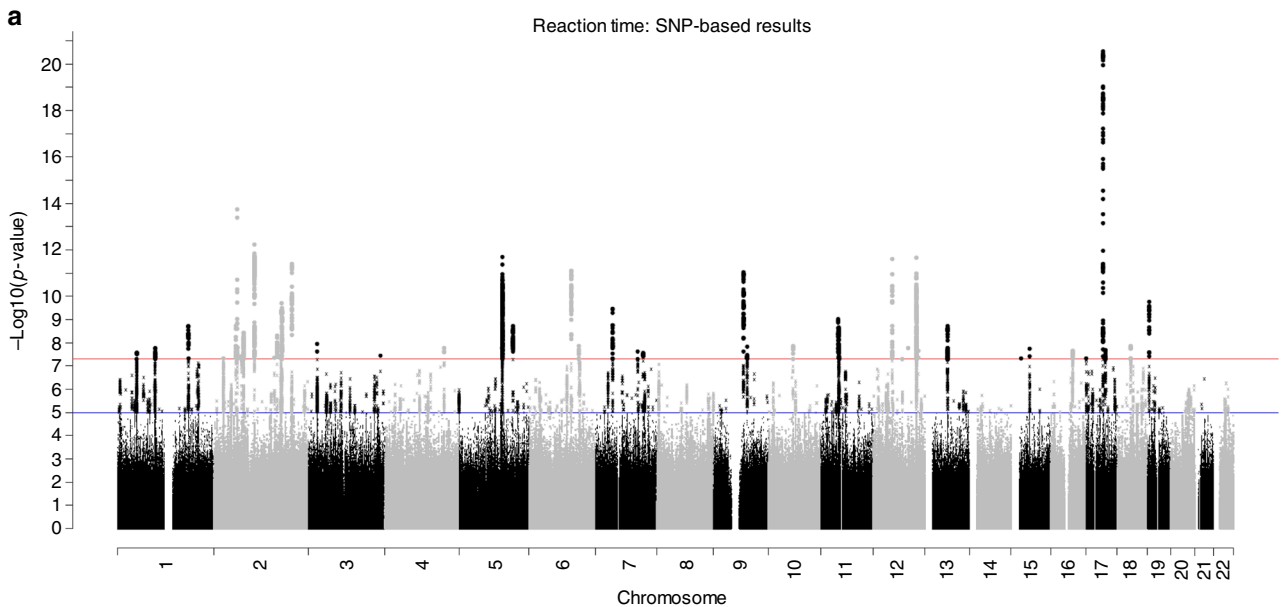

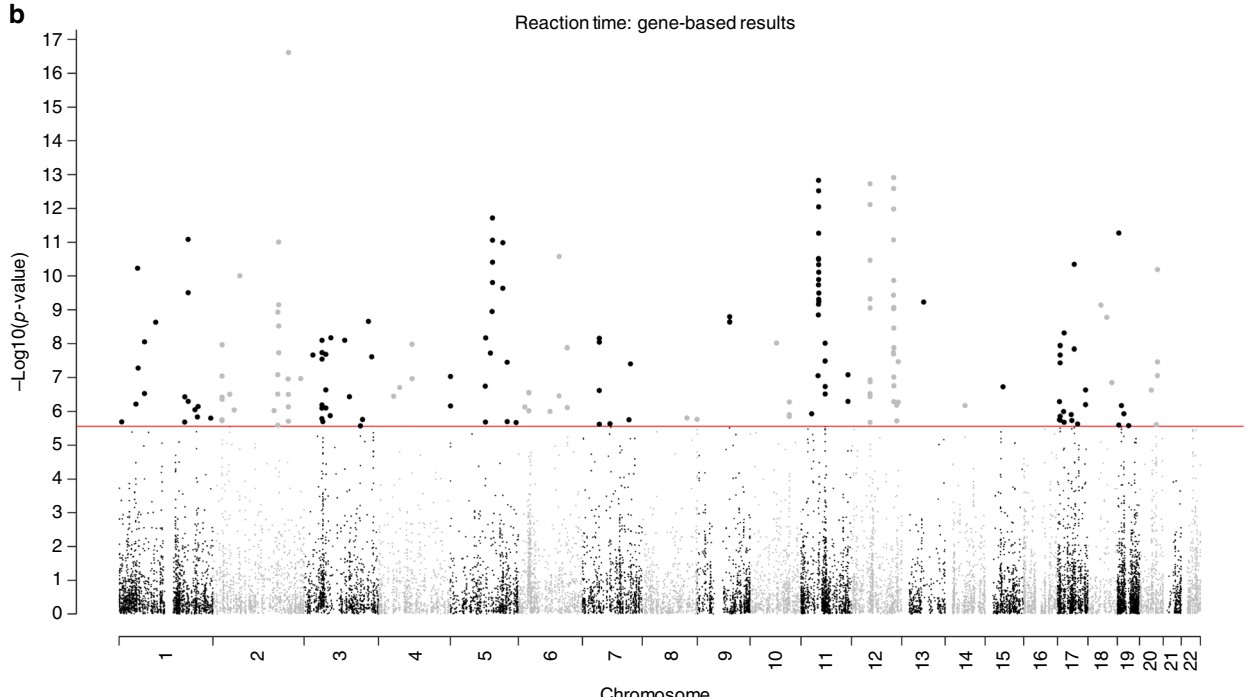

**Fig. 4** Association results for reaction time. SNP-based (**a**) and gene-based (**b**) association results in 330,069 individuals. The red line indicates the threshold for genome-wide significance: $P < 5 \times 10^{-8}$ for (**a**), $P < 2.75 \times 10^{-6}$ for (**b**); the blue line in (**a**) indicates the threshold for suggestive significance: $P < 1 \times 10^{-5}$

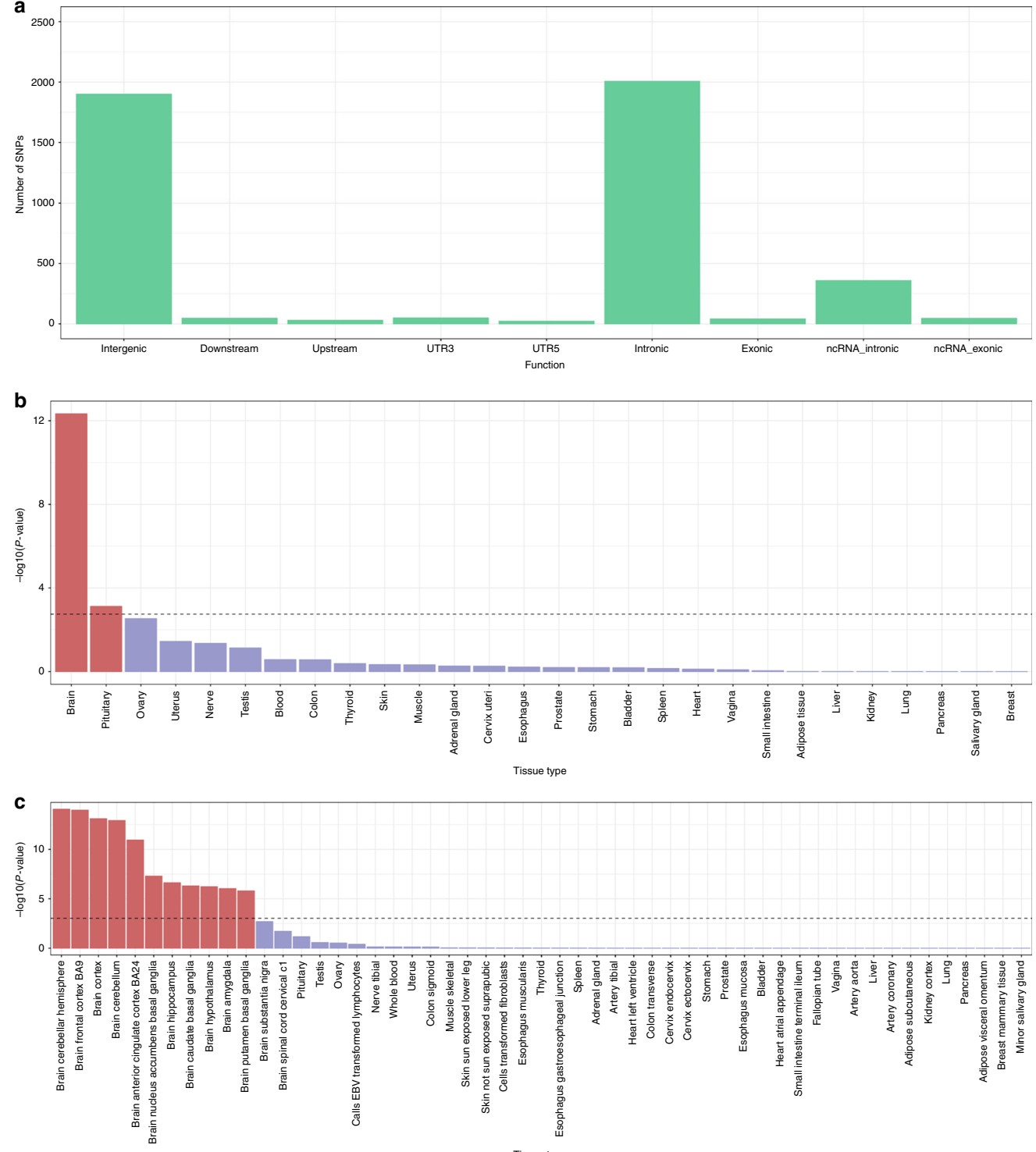

**Fig. 5** Functional analyses of reaction time. Analyses include reaction time-associated SNPs, independent significant SNPs, and all SNPs in LD with independent significant SNPs. Functional consequences of SNPs on genes (**a**) indicated by functional annotation assigned by ANNOVAR. MAGMA gene-property analysis results; results are shown for average expression of 30 general tissue types (**b**) and 53 specific tissue types (**c**). The dotted line indicates the Bonferroni-corrected α level

**Reaction time results**. GWAS results for mean reaction time uncovered 2022 significant SNPs in 42 independent genomic loci (Fig. 4a; Supplementary Fig. 2c; Supplementary Data 10). Suggestive findings are presented in Supplementary Data 11. Both of the significant loci previously reported for this phenotype were replicated[13]. SNPs within the 42 independent genomic loci showed clear evidence of functionality (Fig. 5a; Supplementary Data 12). Using gene-based GWA, a total of 191 genes attained statistical significance (Fig. 4b; Supplementary Fig. 2d; Supplementary Data 13), replicating 18 of the 23 genome-wide

significant genes found previously for this phenotype[13]. Gene-set analysis identified no gene sets associated with reaction time (Supplementary Data 14). Gene-property analysis indicated a role for genes expressed in the brain ($P = 4.66 \times 10^{-13}$), with this link between gene transcription levels and gene-based association with reaction time being found across the cortex (Fig. 5b, c; Supplementary Table 3; Supplementary Data 15). Gene transcription levels observed in the pituitary gland were also linked to gene-based associations with differences in reaction time ($P = 7.60 \times 10^{-4}$).

The SNP-based heritability of reaction time was 7.42% (SE = 0.29). It should be noted that this estimate is likely to be an underestimation due to the method used (LD score regression)[26]. Significant overlap was found between the genetic architecture of reaction time and these health outcomes: ADHD, bipolar disorder, schizophrenia, subjective wellbeing, hand grip strength, sleep duration, maternal longevity, hypertension and neuroticism (Supplementary Data 9). The polygenic score for reaction time explained 0.43% of the general cognitive function variance in ELSA ($P = 1.42 \times 10^{-9}$), 0.56% in Generation Scotland ($P = 2.49 \times 10^{-11}$), and 0.26% in Understanding Society ($P = 1.50 \times 10^{-6}$). The full results for all five thresholds can be found in Supplementary Table 2.

We found a genetic correlation ($r_g$) of 0.247 ($P = 1.28 \times 10^{-30}$) between reaction time and general cognitive function. Overlapping results between the two phenotypes were explored further.

Of the 11,600 genome-wide significant SNPs for general cognitive function, 8269 had a consistent direction of effect with reaction time (sign test, $P = 2.2 \times 10^{-16}$) (Supplementary Data 1). For reaction time, 1070 of the 2022 significant SNPs were consistent for direction of effect with general cognitive function (sign test, $P = 0.0071$) (Supplementary Data 10). One hundred and sixty SNPs were genome-wide significant for both general cognitive function and reaction time, with 82 consistent for direction of effect (sign test, NS) (Supplementary Data 16). These overlapping genome-wide findings are located within six genomic loci (genomic loci: 13, 15, 19, 28, 69, 133; see Supplementary Data 4 for details of loci); two of these are novel loci for general cognitive function. In the gene-based analyses of both the general cognitive function and reaction time phenotypes, there were 39 overlapping significant genes; 13 of these are newly-identified associations with general cognitive function (Supplementary Data 17).

## Discussion

In these meta-analyses of genome-wide association studies for both general cognitive function and reaction time ($N = 300,486$; $N = 330,069$, respectively), we make several original contributions. We report 148 genome-wide significant loci for general cognitive function, of which 58 loci have not been reported before. We report 42 genome-wide significant loci for reaction time, of which 40 have not been reported previously. We also report 291 gene-based associations for general cognitive function, and 173 for reaction time, which have not been reported already. Of these genome-wide significant results, six loci and 39 gene-based associations are genome-wide significant for both general cognitive function and reaction time. We are able to predict, using polygenic scoring, up to 4.31 and 0.56% of the general cognitive function variance in an independent sample, for general cognitive function and reaction time polygenic scores, respectively. We present original and updated estimates of genetic correlations with many health traits for both general cognitive function and reaction time. Gene-set analyses identified significant associations for general cognitive function

with gene-sets involved in neural and cell development. Significant enrichments were observed with genes expressed in the cerebellum and the brain's cortex for both general cognitive function and reaction time.

Upon additional exploration of the 58 newly-associated genetic loci, we find that many contain genes that are of further interest. All of the genes discussed below are also genome-wide significant in the general cognitive function gene-based association analysis ($P < 2.75 \times 10^{-6}$; Supplementary Data 6). Significant gene-based associations with general cognitive function have also been previously reported for *GATAD2B*, *SLC39A1*, and *AUTS2*[16,17].

*GATAD2B* and *SLC39A1* are located on chromosome 1; locus 11. Mutations in *GATAD2B* have been linked to intellectual disability[27]. *SLC39A1* has been implicated in Alzheimer's Disease[28]. The *ATXN1* gene (chromosome 6; locus 60), encodes a protein containing a polyglutamine tract that has previously been associated with Spinocerebellar Ataxia 1[29]. *ATXN1L*, *ATXN2L*, and *ATXN7L2* were also located in significant loci that have previously been associated with cognitive function, intelligence, or educational attainment[16,17,24]. The *DCDC2* gene (chromosome 6; locus 64) has previously been associated with cortical morphology[30], dyslexia[31], and normal variation in reading and spelling[32], but not with general cognitive function. *TTBK1* (chromosome 6; locus 66) encodes a neuron-specific serine/threonine and tyrosine kinase, which regulates phosphorylation of tau[33]. Genetic variants in this gene have been associated with Alzheimer's disease[34]. *AUTS2* (chromosome 7; locus 72) is implicated in a number of neurological disorders[35]. Mutations in *CWF19L1* (chromosome 10; locus 91) have been associated with spinocerebellar ataxia and intellectual disability[36]. *RBFOX1* (chromosome 16; locus 121) encodes a mRNA-splicing factor that interacts with *ATXN2*[37], and mutations in this gene lead to neurodevelopmental disorders[38]. Locus 131, on chromosome 17, has previously been associated with Smith-Magenis Syndrome[39]. The most significantly-associated SNP ($P = 2.2 \times 10^{-8}$) in this locus lies in an intron of the *RAI1* gene. *RAI1* encodes a protein containing a polymorphic polyglutamine tract that is expressed mainly in neuronal tissues. Variants in the gene are also associated with schizophrenia[40].

Of the seven significant gene sets identified, one was a new finding: 'positive regulation of nervous system development'. A more detailed description of this gene-set is: 'any process that activates, maintains or increases the frequency, rate or extent of nervous system development, the origin and formation of nervous tissue'. The remaining six gene-sets showed replication with previous studies of general cognitive function and/or education[16,17,24]. Only one, 'regulation of cell development', was significant across all four studies[16,17,24]. Identification of these gene sets is consistent with genes associated with cognitive function regulating the generation of cells within the nervous system, including the formation of neuronal dendrites.

A number of not-previously-reported genetic correlations with cognitive function were found here, including with cardiovascular variables. For example, it is already known that there is a phenotypic association between cognitive function in youth and the development of hypertension by age 50 years[41]; we found a genetic correlation of −0.15. Other genetic correlations between cardiovascular variables and cognitive function were angina ($r_g = -0.18$) and heart attack ($r_g = -0.17$); again, there are known to be phenotypic associations between prior cognitive functioning and various cardiovascular outcomes[41,42].

The genetic correlations between general cognitive function and eyesight were in opposite directions depending on the reported reason for wearing glasses or contact lenses; this was despite an

overall positive genetic correlation between general cognitive function and wearing glasses ($r_g = 0.28$). The result for myopia (short-sightedness; $r_g = 0.32$) was consistent with previous evidence of a positive phenotypic[43] and genetic[44] correlation between this trait and cognitive function. Less genetic work has investigated the links between hyperopia (long-sightedness) and cognitive function, although our finding, a genetic correlation of $r_g = -0.21$, was consistent with the negative phenotypic association between these variables reported in previous literature[45].

We have investigated the six regions of the genome identified as having a shared effect between general cognitive function and more elementary cognitive tasks. Locus 13 on chromosome 1 contains the *NMNAT2* gene. *NMNAT2* is involved with Wallerian degeneration[46,47]; this is a neurodegenerative process which occurs after axonal injury in both the peripheral and central nervous system. Locus 15 on chromosome 2 contains *ENSG00000271894*, a non-coding RNA gene. *SLC4A10* and *DPP4* are located on chromosome 2 (locus 28). Variants in both *SLC4A10* and *DPP4* have been linked to schizophrenia[48,49]; hippocampal volume has also been linked to variants in *DPP4*[50]. A variant of *FOXO3* (chromosome 6; locus 69) has been shown to be associated with longevity in humans[51,52]; it is found in most centenarians across a variety of populations. *MAPT, WNT3, CRHR1, KANSL1*, and *NSF* are located on chromosome 17, locus 133; genetic variants within these genes have been linked to Alzheimer's disease in *APOE* e4 carriers[53], Parkinson's disease[54–56], neuroticism[57], infant head circumference[58], intracranial volume[59], and subcortical brain region volumes[60]. Researchers following up the present study's results could prioritise the genetic loci uncovered herein that are associated with general cognitive function and reaction time (Supplementary Data 16 and 17), as well as those that are also associated with brain-related measures in other large GWASs. Such variants, being associated with multiple cognitive and neurological phenotypes, might help to prioritise potentially causal variants, and help to identify how differences in genotypic sequence are linked to such phenotypic consequences.

We note limitations with the cognitive phenotypes studied. For general cognitive function, phenotypic heterogeneity is a limitation, due to different tests being used in most samples. We also note the small number of cognitive tests being used in the construction of the general cognitive function phenotype in some cohorts. However, we were able to investigate this further by estimating genetic correlations for general cognitive function amongst some of the larger cohorts. These demonstrated strong positive genetic correlations that ranged from $r_g = 0.88$–$1.0$ (Table 2). There were slight differences in the test questions and the testing environment for the UK Biobank's 'fluid' (verbal-numerical reasoning) test in the assessment centre versus the online version. We used a bivariate GREML analysis to investigate the genetic contribution to the stability of individual differences in people's verbal-numerical reasoning; we report a significant perfect genetic correlation. The UK Biobank's reaction time variable is based on only four trials per participant; this is far fewer trials than would typically be measured. For example, other large UK surveys have used 40 trials in choice RT procedures[61,62].

Both the overall size of the present study's meta-analysis of GWASs and the inclusion of a single large sample, UK Biobank, are strengths, which contributed to the abundance of new findings. When compared to an analysis of only UK Biobank herein, the current meta-analysis adds 92 independent significant loci, 51 of which are novel. Yet, as genome-wide studies of other complex traits continue to increase up to and beyond a million individuals, an even larger sample size will be required in order to seek replication of these findings,

identify new associations, and generate stronger polygenic predictions[15,63] (Supplementary Fig. 1).

When compared to previous large studies of cognitive function and education, we replicate a large proportion, but not all, of the previously-reported significant findings. These differences in reported findings might be explained partly by differences in study populations (including age, social status, and ethnicity), phenotypes, and analysis methods. Whereas we know that there is sample overlap in the studies described, each comprises a unique set of contributing cohorts. As described above, there is substantial variation in the cognitive tests that contribute to the construction of a general cognitive function phenotype. Cognitive function is not as simple to measure as, say, height, and it is far from being standardised. This limitation applies across the GWAS meta-analysis studies, as well as within them. The use of different analysis methods—for example MTAG, which includes phenotypes other than the target phenotype—might also contribute to the different findings that have been reported. Finally, it is also possible that, although specific loci reached genome-wide significance in particular studies, there are false positives, highlighting the importance of well-powered replication studies.

Gene-based analysis has been shown to increase the power to detect associations, because the multiple testing burden is reduced, and the effects of multiple SNPs are combined together. From these gene-based analyses, the association of a gene with general cognitive function does not imply that it is causally related to this phenotype, only that the gene is in a region of strong association within a locus. These loci may contain multiple associated genes; therefore, we note that all of the associated genes that we reported may not be independent findings. However, we note that gene-based testing will not be able to detect associations that fall outside of the gene-body. This means that, if SNPs in promoter regions harbour variants that are causal to differences in general cognitive function or reaction time, they will be missed in our gene-based analyses.

General cognitive function has prominence and pervasiveness in the human life course, and it is important to understand the environmental and genetic origins of its variation in the population[4]. The unveiling here of many genetic loci, genes, and genetic pathways that contribute to its heritability (Fig. 2; Supplementary Data 1, 6 and 7)—which it shares, as we find here, with many health outcomes, longevity, brain structure, and processing speed—provides a foundation for exploring the mechanisms that bring about and sustain cognitive efficiency through life.

## Methods

**Participants and cognitive phenotypes**. The present study includes 300,486 individuals of European ancestry from 57 population-based cohorts brought together by the Cohorts for Heart and Aging Research in Genomic Epidemiology (CHARGE), the Cognitive Genomics Consortium (COGENT) consortia, and UK Biobank (Supplementary Note 2). All individuals were aged between 16 and 102 years. Exclusion criteria included clinical stroke (including self-reported stroke) or prevalent dementia (Supplementary Data 18).

General cognitive function, unlike height for example, is not measured the same way in all samples. Here, this was mitigated by applying a consistent method of extracting a general cognitive function component from cognitive test data in the cohorts of the CHARGE and COGENT consortia; all individuals were of European ancestry (Supplementary Note 1).

For each of the CHARGE and COGENT cohorts, a general cognitive function component phenotype was constructed from a number of cognitive tasks. Each cohort was required to have tasks that tested at least three different cognitive domains. We avoided taking more than one cognitive test score from any individual cognitive test. Principal component analysis was applied to the cognitive test scores to derive a measure of general cognitive function. Principal component analyses results for the CHARGE cohorts were checked by one author (IJD) to establish the presence of a single component. The scree slope was examined, the percentage of variance accounted for by the first unrotated principal component was noted, and it was checked that all tests had sufficient loading on the first

unrotated principal component. Scores on the first unrotated component were used as the cognitive phenotype (general cognitive function). Principal component analyses for the COGENT cohorts are described in Trampush et al. (pp. 337–338, and Supplementary Table 1)[64].

UK Biobank participants were asked 13 multiple-choice questions that assessed verbal and numerical reasoning (VNR: UK Biobank calls this the 'fluid' cognitive test). The VNR score was the number of questions answered correctly in 2 min. Four samples of UK Biobank participants with verbal-numerical reasoning scores were used in the current analyses. The first sample (VNR Assessment Centre) consists of UK Biobank participants who completed the verbal-numerical reasoning test at baseline in assessment centres ($n = 107,586$). The second UK Biobank sample (VNR T2) consists of participants who did not complete the verbal-numerical reasoning test at baseline but did complete this test at the first repeat assessment visit in assessment centres ($n = 11,123$). The third UK Biobank sample (VNR MRI) consists of participants who did not complete the verbal-numerical reasoning test at a previous testing occasion but did complete the test at the imaging visit in assessment centres ($n = 3002$). The fourth UK Biobank sample (VNR Web-Based) consists of participants who did not complete the verbal-numerical reasoning test at any assessment centre visit, but did complete this test during the web-based cognitive assessment online ($n = 46,322$). Details of the cognitive phenotypes for all cohorts can be found in Supplementary Note 1.

At the baseline UK Biobank assessment, 496,790 participants completed the reaction time test. Details of the test can be found in Supplementary Note 1. A sample of 330,069 UK Biobank participants with scores on both the reaction time test and genotyping data was used in this study.

**Genome-wide association analyses**. Genotype–phenotype association analyses were performed within each cohort, using an additive model, on imputed SNP dosage scores. Adjustments for age, sex, and population stratification were included in the model for each cohort. Cohort-specific covariates—for example, site or familial relationships—were also fitted as required. Cohort-specific quality control procedures, imputation methods, and covariates are described in Supplementary Data 19. Quality control of the cohort-level summary statistics was performed using the EasyQC software[65], which implemented the exclusion of SNPs with imputation quality <0.6 and minor allele count <25.

**General cognitive function meta-analysis**. A meta-analysis including all the CHARGE-COGENT and UK Biobank summary results was performed using the METAL package with a sample-size weighted model implemented (http://www.sph.umich.edu/csg/abecasis/Metal).

**Reaction time genome-wide association analysis**. The GWAS of reaction time from the UK Biobank sample was performed using the BGENIE v1.2 analysis package (https://jmarchini.org/bgenie/). A linear SNP association model was tested which accounted for genotype uncertainty. Reaction time was adjusted for the following covariates: age, sex, genotyping batch, genotyping array, assessment centre, and 40 principal components.

**Genomic risk loci characterization using FUMA**. Genomic risk loci were defined from the SNP-based association results, using FUnctional Mapping and Annotation of genetic associations (FUMA)[23]. Firstly, independent significant SNPs were identified using the SNP2GENE function and defined as SNPs with a $P$-value of $\leq 5 \times 10^{-8}$ and independent of other genome wide significant SNPs at $r^2 < 0.6$. Using these independent significant SNPs, tagged SNPs to be used in subsequent annotations were identified as all SNPs that had a MAF $\geq 0.0005$ and were in LD of $r^2 \geq 0.6$ with at least one of the independent significant SNPs. These tagged SNPs included those from the 1000 genomes reference panel and need not have been included in the GWAS performed in the current study. Genomic risk loci that were 250 kb or closer were merged into a single locus. Lead SNPs were also identified using the independent significant SNPs and were defined as those that were independent from each other at $r^2 < 0.1$.

**Comparison with previous findings**. Previous evidence of association for each of the 148 genetic loci identified herein as being associated with general cognitive function was sought in the largest published GWASs of general cognitive function[16,17] and education[24]. We performed look-ups on all tagged SNPs ($r^2 > 0.6$) within each locus, including all 1000 genomes SNPs, and classed any tagged SNP previously reported as genome-wide significant, as replication. Details of these findings are presented in Supplementary Data 3.

**Gene-based analysis implemented in FUMA**. Gene-based analysis has been shown to increase the power to detect genotype-phenotype association because the multiple testing burden is reduced, and the effect of multiple SNPs is combined together[66]. Gene-based analysis was conducted using MAGMA[67]. The test carried out using MAGMA, as implemented in FUMA, was the default SNP-wise test using the mean $\chi^2$ statistic derived on a per gene basis. SNPs were mapped to genes based on genomic location. All SNPs that were located within the gene-body were used to derive a $P$-value describing the association found with general cognitive function

and reaction time. The SNP-wise model from MAGMA was used and the NCBI build 37 was used to determine the location and boundaries of 18,199 autosomal genes. Linkage disequilibrium within and between each gene was gauged using the 1000 genomes phase 3 release[68]. A Bonferroni correction was applied to control for multiple testing; the genome-wide significance threshold was $P < 2.75 \times 10^{-6}$.

**Estimation of SNP-based heritability**. The proportion of variance explained by all common SNPs was estimated using univariate GCTA-GREML analyses[69] in four of the largest individual cohorts: ELSA, Understanding Society, UK Biobank, and Generation Scotland. Sample sizes for all of the GCTA analyses in these cohorts differed from the association analyses, because one individual was excluded from any pair of individuals who had an estimated coefficient of relatedness of >0.025 to ensure that effects due to shared environment were not included. The same covariates were included in all GCTA-GREML analyses as for the SNP-based association analyses.

**Univariate Linkage Disequilibrium Score regression**. Univariate LDSC regression was performed on the summary statistics from the GWAS on general cognitive function and reaction time. The heritability $Z$-score provides a measure of the polygenic signal found in each data set. Values greater than four indicate that the data are suitable for use with bivariate LDSC regression[70]. The mean $\chi^2$ statistic indicates the inflation of the GWAS test statistics that, under the null hypothesis of no association (i.e., no inflation of test statistics), would be one. An inflation in the test statistics can indicate population stratification, cryptic relatedness, or the presence of many alleles each with a small effect. The intercept of the LDSC regression can detect the difference between inflation due to stratification and cryptic relatedness, and the inflation due to a polygenic signal. This is because the inflation in test statistics attributable to stratification, drift, and cryptic relatedness will not correlate with LD, whereas inflation due to polygenicity will. The LDSC regression intercept, therefore, captures the inflation in the $\chi^2$ statistics that is not due to stratification or other confounds.

For each GWAS, an LD regression was carried out by regressing the GWA test statistics ($\chi^2$) on to each SNP's LD score, which is the sum of squared correlations between the minor allele frequency count of a SNP with the minor allele frequency count of every other SNP. This regression allows for the estimation of heritability from the slope, and a means to detect residual confounders using the intercept. For general cognitive function, we report an LD score regression intercept of 1.058 (SE = 0.011) and a ratio of 0.0659; this indicates that only 6.6% of the inflation observed can be ascribed to causes other than a polygenic signal. For reaction time, we report an LD score regression intercept of 1.02 (SE = 0.009) and a ratio 0.0475; this indicates that only 4.75% of the inflation observed can be ascribed to causes other than a polygenic signal.

LD scores and weights were downloaded from (http://www.broadinstitute.org/~bulik/eur_ldscores/) for use with European populations. A minor allele frequency cut-off of >0.1 and an imputation quality score of >0.9 were applied to the GWAS summary statistics. Following this, SNPs were retained if they were found in HapMap 3 with MAF >0.05 in the 1000 Genomes EUR reference sample. Following this, indels and structural variants were removed along with strand ambiguous variants. SNPs whose alleles did not match those in the 1000 Genomes were also removed. As the presence of outliers can increase the standard error in LDSC score regression[70] and so SNPs where $\chi^2 > 80$ were also removed.

**Genetic correlations**. Genetic correlations were estimated using two methods, bivariate GCTA-GREML[71] and LDSC[70]. Bivariate GCTA was used to calculate genetic correlations between phenotypes and cohorts where the genotyping data were available. This method was used to calculate the genetic correlations between different cohorts for the general cognitive function phenotype. It was also employed to investigate the genetic contribution to the stability of the same UK Biobank's participants' verbal-numerical reasoning test scores in the assessment centre and then in web-based, online testing. In cases where only GWA summary results were available, bivariate LDSC was used to estimate genetic correlations between two traits. This was used to estimate the degree of overlap between polygenic architecture of the traits. Bivariate LDSC regression was used to estimate genetic correlations between general cognitive function, reaction time, and the following health outcomes: ADHD, age at menarche, age at menopause, Alzheimer's disease, anorexia nervosa, bipolar disorder, BMI, bone density femoral neck, bone density lumbar spine, coronary artery disease, HbA1c, HDL cholesterol, hippocampal volume, intracranial volume, LDL cholesterol, longevity, lung cancer, major depression, neuroticism, schizophrenia, smoking status, triglycerides, type 2 diabetes, waist-hip ratio, autism spectrum disorder, birth weight, depressive symptoms, hypertension, pulse wave arterial stiffness, angina, heart attack, parental longevity, forced expiratory volume in 1-second (FEV1), hand grip strength, happiness, health satisfaction, heel bone mineral density, osteoarthritis, overall health rating, wearing of glasses or contact lenses, long-sightedness, short-sightedness, sleep duration, sleeplessness/insomnia, and subjective wellbeing. For Alzheimer's disease, a 500-kb region surrounding APOE was excluded and the analysis re-run (Alzheimer's disease (500 kb)). Supplementary Data 20 provides further details on the sources of the GWAS summary statistics.

**Polygenic prediction**. Polygenic profile score analyses were used to predict cognitive test performance in Generation Scotland, the English Longitudinal Study of Ageing, and Understanding Society. Polygenic profiles were created in PRSice[72] using results of a general cognitive function meta-analysis that excluded the Generation Scotland, the English Longitudinal Study of Ageing, and Understanding Society cohorts. Polygenic profiles were also created in these cohorts based on the UK Biobank GWA reaction time results. SNPs with a MAF < 0.01 were removed prior to creating the polygenic profiles. Clumping was used to obtain SNPs in linkage disequilibrium with an $r^2 < 0.25$ within a 250 kb window. Polygenic profile scores were created at $P$-value thresholds of 0.01, 0.05, 0.1, 0.5, and 1 (all SNPs), based on the significance of the association in the general cognitive function and reaction time GWAS. Linear regression models were used to examine the associations between the polygenic profile and cognitive ability in GS, ELSA, and US, adjusting for age at measurement, sex, and the first 10 (GS), 15 (ELSA), and 20 (US) genetic principal components to adjust for population stratification. The false discovery rate (FDR) method was used to correct for multiple testing across the polygenic profiles at all five thresholds[73].

**Functional annotation implemented in FUMA[23]**. The independent significant SNPs and those in LD with the independent significant SNPs were annotated for functional consequences on gene functions using ANNOVAR[74] and the Ensembl genes build 85. A CADD score[75], RegulomeDB score[76], and 15-core chromatin states[77–79] were obtained for each SNP. eQTL information was obtained from the following databases: GTEx (http://www.gtexportal.org/home/), BRAINEAC (http://www.braineac.org/), Blood eQTL Browser (http://genenetwork.nl/bloodeqtlbrowser/), and BIOS QTL browser (http://genenetwork.nl/biosqtlbrowser/). Functionally-annotated SNPs were then mapped to genes based on physical position on the genome, eQTL associations (all tissues) and chromatin interaction mapping (all tissues). Intergenic SNPs were mapped to the two closest up- and down-stream genes which can result in their being assigned to multiple genes.

**Gene-set analysis implemented in FUMA**. In order to test whether the polygenic signal measured in each of the GWASs clustered in specific biological pathways, a competitive gene-set analysis was performed. Gene-set analysis was conducted in MAGMA[67] using competitive testing, which examines if genes within the gene set are more strongly associated with each of the cognitive phenotypes than other genes. Such competitive tests have been shown to control for Type 1 error rate as well as facilitating an understanding of the underlying biology of cognitive differences[80,81]. A total of 10,891 gene-sets (sourced from Gene Ontology[82], Reactome[83], and, SigDB[84]) were examined for enrichment of general cognitive function and reaction time. A Bonferroni correction was applied to control for the multiple tests performed on the 10,891 gene sets available for analysis.

**Gene-property analysis implemented in FUMA**. A gene-property analysis was conducted using MAGMA in order to indicate the role of particular tissue types that influence differences in general cognitive function and reaction time. The goal of this analysis was to test if, in 30 broad tissue types and 53 specific tissues, tissue-specific differential expression levels were predictive of the association of a gene with general cognitive function and reaction time. Tissue types were taken from the GTEx v6 RNA-seq database[85] with expression values being log2 transformed with a pseudocount of 1 after winsorising at 50, with the average expression value being taken from each tissue. Multiple testing was controlled for using a Bonferroni correction.

**Data availability**. The GWAS summary results for all significant and suggestive SNPs for general cognitive function and reaction time are available in Supplementary Data 1, 2, 10 and 11. The full GWAS summary results for Reaction Time are available to download here: http://www.ccace.ed.ac.uk/node/335. Access to the full GWAS summary results for general cognitive function can be requested by application to the chairs of the CHARGE and COGENT consortia.

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

## Acknowledgements

This research was conducted in The University of Edinburgh Centre for Cognitive Ageing and Cognitive Epidemiology, funded by the Biotechnology and Biological Sciences Research Council and Medical Research Council (MR/K026992/1). This research was conducted using the UK Biobank Resource (Application Nos. 10279 and 4844). The Neurology Working Group within the Cohorts for Heart and Aging Research in Genomic Epidemiology is partly supported by grants from the National Institute on Aging (R01 AG033193, U01 AG049505 and U01 AG052409). Cohort-specific acknowledgements are in Supplementary Note 3.

## Author Contributions

G.D. and I.J.D. drafted the manuscript with contributions from M.Lu, S.E.H., W.D.H., S. J.R., S.P.H., C.F-R., and J.O.; G.D., J.W.T. and M.La performed quality control of the CHARGE-COGENT data. I.J.D. designed and supervised the cognitive psychometric analyses in the CHARGE cohorts. G.D. and R.E.M. performed quality control of UK Biobank data. G.D., J.W.T. and M.La analysed the data. S.E.H., W.D.H., S.P.H., and M.Lu performed/assisted with downstream analysis. G.D. and I.J.D. co-ordinated the CHARGE and UK Biobank work, and their integration with COGENT; T.L., J.W.T. and M.La co-ordinated the COGENT work. All authors supplied phenotype data, genotype data, and GWA results, and commented on and approved the manuscript.

## Additional information

**Competing interests:** A.M.D. is a Founder of and holds equity in CorTechs Labs, Inc., and serves on its Scientific Advisory Board. He is a member of the Scientific Advisory Board of Human Longevity, Inc., and receives funding through research agreements with General Electric Healthcare and Medtronic, Inc. The terms of these arrangements have been reviewed and approved by UCSD in accordance with its conflict of interest policies. B.M.P. serves on a DSMB for a clinical trial of a device funded by the manufacturer (Zoll LifeCor), and on the steering committee of the Yale Open Data Access Project funded by Johnson & Johnson. I.J.D. is a participant in UK Biobank. All other authors declare no competing interests.

Gail Davies[1], Max Lam[2], Sarah E. Harris[1,3], Joey W. Trampush[4,5], Michelle Luciano[1], W. David Hill[1], Saskia P. Hagenaars[1,6], Stuart J. Ritchie[1], Riccardo E. Marioni[1,3], Chloe Fawns-Ritchie[1], David C.M. Liewald[1], Judith A. Okely[1], Ari V. Ahola-Olli[7,8], Catriona L.K. Barnes[9], Lars Bertram[10], Joshua C. Bis[11], Katherine E. Burdick[12,13,14], Andrea Christoforou[15,16], Pamela DeRosse[2,17], Srdjan Djurovic[15,18], Thomas Espeseth[19,20], Stella Giakoumaki[21], Sudheer Giddaluru[15,16], Daniel E. Gustavson[22,23], Caroline Hayward[24,25], Edith Hofer[26,27], M. Arfan Ikram[28,29,30], Robert Karlsson[31], Emma Knowles[32], Jari Lahti[33,34], Markus Leber[35], Shuo Li[36], Karen A. Mather[37], Ingrid Melle[15,19], Derek Morris[38], Christopher Oldmeadow[39], Teemu Palviainen[40], Antony Payton[41], Raha Pazoki[42], Katja Petrovic[26], Chandra A. Reynolds[43], Muralidharan Sargurupremraj[44], Markus Scholz[45,46], Jennifer A. Smith[47,48], Albert V. Smith[49,50], Natalie Terzikhan[28,51], Anbupalam Thalamuthu[37], Stella Trompet[52], Sven J. van der Lee[28], Erin B. Ware[48], B. Gwen Windham[53], Margaret J. Wright[54,55], Jingyun Yang[56,57], Jin Yu[17], David Ames[58,59], Najaf Amin[28], Philippe Amouyel[60], Ole A. Andreassen[19,61], Nicola J. Armstrong[62], Amelia A. Assareh[37], John R. Attia[63], Deborah Attix[64,65], Dimitrios Avramopoulos[66,67], David A. Bennett[56,57], Anne C. Böhmer[68,69], Patricia A. Boyle[56,70], Henry Brodaty[37,71], Harry Campbell[9], Tyrone D. Cannon[72], Elizabeth T. Cirulli[73], Eliza Congdon[74], Emily Drabant Conley[75], Janie Corley[1], Simon R. Cox[1], Anders M. Dale[22,76,77,78], Abbas Dehghan[42,79], Danielle Dick[80], Dwight Dickinson[81], Johan G. Eriksson[82,83,84,85], Evangelos Evangelou[42,82], Jessica D. Faul[48], Ian Ford[86], Nelson A. Freimer[74], He Gao[42], Ina Giegling[87], Nathan A. Gillespie[88], Scott D. Gordon[89], Rebecca F. Gottesman[90,91], Michael E. Griswold[92], Vilmundur Gudnason[49,50], Tamara B. Harris[93], Annette M. Hartmann[87], Alex Hatzimanolis[94,95,96], Gerardo Heiss[97], Elizabeth G. Holliday[63], Peter K. Joshi[9], Mika Kähönen[98,99,100], Sharon L.R. Kardia[47], Ida Karlsson[31], Luca Kleineidam[101,102,103,104], David S. Knopman[105], Nicole A. Kochan[37,106], Bettina Konte[87], John B. Kwok[107,108], Stephanie Le Hellard[15,16], Teresa Lee[37,106], Terho Lehtimäki[109,110], Shu-Chen Li[111,112], Christina M. Lill[113], Tian Liu[10,111], Marisa Koini[26], Edythe London[74], Will T. LongstrethJr[114,115], Oscar L. Lopez[116], Anu Loukola[40], Tobias Luck[46,117], Astri J. Lundervold[118,119], Anders Lundquist[120,121], Leo-Pekka Lyytikäinen[109,110], Nicholas G. Martin[89], Grant W. Montgomery[89,122], Alison D. Murray[25,123], Anna C. Need[124], Raymond Noordam[52], Lars Nyberg[120,125,126], William Ollier[127], Goran Papenberg[111,128], Alison Pattie[129], Ozren Polasek[130,131], Russell A. Poldrack[132], Bruce M. Psaty[11,133,134], Simone Reppermund[37,135], Steffi G. Riedel-Heller[117], Richard J. Rose[136], Jerome I. Rotter[137,138], Panos Roussos[12,139,140], Suvi P. Rovio[7], Yasaman Saba[141], Fred W. Sabb[142], Perminder S. Sachdev[37,106], Claudia L. Satizabal[143,144], Matthias Schmid[145], Rodney J. Scott[63], Matthew A. Scult[146], Jeannette Simino[92], P. Eline Slagboom[147], Nikolaos Smyrnis[94,95], Aïcha Soumaré[44], Nikos C. Stefanis[94,95,96], David J. Stott[148], Richard E. Straub[149],

Kjetil Sundet[19,20], Adele M. Taylor[129], Kent D. Taylor[137,138], Ioanna Tzoulaki[42,79,150], Christophe Tzourio[44,151], André Uitterlinden[28,152], Veronique Vitart[24], Aristotle N. Voineskos[153], Jaakko Kaprio[40,82,154], Michael Wagner[103,104], Holger Wagner[102], Leonie Weinhold[145], K. Hoyan Wen[28], Elisabeth Widen[40], Qiong Yang[36], Wei Zhao[47], Hieab H.H. Adams[28,155], Dan E. Arking[67], Robert M. Bilder[74], Panos Bitsios[156], Eric Boerwinkle[157,158], Ornit Chiba-Falek[64], Aiden Corvin[159], Philip L. De Jager[160,161], Stéphanie Debette[44,162], Gary Donohoe[38], Paul Elliott[42,79], Annette L. Fitzpatrick[115,163], Michael Gill[159], David C. Glahn[32], Sara Hägg[31], Narelle K. Hansell[54], Ahmad R. Hariri[146], M. Kamran Ikram[28,30], J. Wouter Jukema[164], Eero Vuoksimaa[40,154], Matthew C. Keller[165], William S. Kremen[22,23], Lenore Launer[93], Ulman Lindenberger[111], Aarno Palotie[40,166,167], Nancy L. Pedersen[31], Neil Pendleton[168], David J. Porteous[1,3,25], Katri Räikkönen[33], Olli T. Raitakari[7,169], Alfredo Ramirez[35,68,102], Ivar Reinvang[20], Igor Rudan[9], Dan Rujescu[87], Reinhold Schmidt[26], Helena Schmidt[141], Peter W. Schofield[170], Peter R. Schofield[171,172], John M. Starr[1,173], Vidar M. Steen[15,16], Julian N. Trollor[37,135], Steven T. Turner[174], Cornelia M. Van Duijn[28], Arno Villringer[175,176], Daniel R. Weinberger[149], David R. Weir[48], James F. Wilson[9,24], Anil Malhotra[17,177,178], Andrew M. McIntosh[1,179], Catharine R. Gale[1,180], Sudha Seshadri[142,143,181], Thomas H. MosleyJr[53], Jan Bressler[157], Todd Lencz[17,179] & Ian J. Deary[1]

[1]Centre for Cognitive Ageing and Cognitive Epidemiology, Department of Psychology, School of Philosophy, Psychology and Language Sciences, The University of Edinburgh, Edinburgh EH8 9JZ, UK. [2]Institute of Mental Health, Singapore 539747, Singapore. [3]Medical Genetics Section, Centre for Genomic & Experimental Medicine, MRC Institute of Genetics & Molecular Medicine, University of Edinburgh, Western General Hospital, Edinburgh EH4 2XU, UK. [4]BrainWorkup, LLC, Los Angeles 90033 CA, USA. [5]Department of Psychiatry and Behavioral Sciences, Keck School of Medicine, University of Southern California, Los Angeles 90033 CA, USA. [6]Social, Genetic and Developmental Psychiatry Centre, Institute of Psychiatry, Psychology & Neuroscience, King's College London, De Crespigny Park, Denmark Hill, London SE5 8AF, UK. [7]Research Centre of Applied and Preventive Cardiovascular Medicine, University of Turku, Turku 20520, Finland. [8]Department of Internal Medicine, Satakunta Central Hospital, Pori 28100, Finland. [9]Centre for Global Health Research, Usher Institute of Population Health Sciences and Informatics, University of Edinburgh, Teviot Place, Edinburgh EH8 9AG, Scotland. [10]Max Planck Institute for Molecular Genetics, Berlin 14195, Germany. [11]Cardiovascular Health Research Unit, Department of Medicine, University of Washington, Seattle 98101 Washington, USA. [12]Department of Psychiatry, Icahn School of Medicine at Mount Sinai, New York 10029 NY, USA. [13]Mental Illness Research, Education, and Clinical Center (VISN 3), James J. Peters VA Medical Center, Bronx 10468 NY, USA. [14]Department of Psychiatry, Brigham and Women's Hospital, Harvard Medical School, Boston 02115 MA, USA. [15]NORMENT, K.G. Jebsen Centre for Psychosis Research, Department of Clinical Science, University of Bergen, Bergen 5021, Norway. [16]Dr. Einar Martens Research Group for Biological Psychiatry, Center for Medical Genetics and Molecular Medicine, Haukeland University Hospital, Bergen 5020, Norway. [17]Center for Psychiatric Neuroscience, Feinstein Institute for Medical Research, Manhasset 11030 NY, USA. [18]Department of Medical Genetics, Oslo University Hospital, University of Bergen, Oslo 0424, Norway. [19]Department of Psychology, University of Oslo, Oslo 0373, Norway. [20]Division of Mental Health and Addiction, Oslo University Hospital, Oslo 0315, Norway. [21]Department of Psychology, University of Crete, Crete GR-74100, Greece. [22]Department of Psychiatry, University of California, San Diego 92093 CA, USA. [23]Center for Behavior Genetics of Aging, University of California, San Diego 92093 CA, USA. [24]Medical Research Council Human Genetics Unit, Institute of Genetics and Molecular Medicine, University of Edinburgh, Edinburgh EH4 2XU, UK. [25]Generation Scotland, Centre for Genomic and Experimental Medicine, University of Edinburgh, Edinburgh EH4 2XU, UK. [26]Department of Neurology, Clinical Division of Neurogeriatrics, Medical University of Graz, Graz 8036, Austria. [27]Institute of Medical Informatics Statistics and Documentation, Medical University of Graz, Graz 8036, Austria. [28]Department of Epidemiology, Erasmus University Medical Center, Rotterdam 3015, The Netherlands. [29]Department of Radiology and Nuclear Medicine, Erasmus University Medical Center, Rotterdam 3015, The Netherlands. [30]Department of Neurology, Erasmus University Medical Center, Rotterdam xxxxxx, The Netherlands. [31]Department of Medical Epidemiology and Biostatistics, Karolinska Institutet, Stockholm SE-171 77, Sweden. [32]Department of Psychiatry, Yale University School of Medicine, New Haven 06511 CT, USA. [33]Department of Psychology and Logopedics, Faculty of Medicine, University of Helsinki, Helsinki 00014, Finland. [34]Helsinki Collegium for Advanced Studies, University of Helsinki, Helsinki 00014, Finland. [35]Department of Psychiatry and Psychotherapy, University of Cologne, Cologne D-50937, Germany. [36]Department of Biostatistics, Boston University School of Public Health, Boston 02118 MA, USA. [37]Centre for Healthy Brain Ageing, School of Psychiatry, University of New South Wales, Sydney 2031, Australia. [38]Neuroimaging, Cognition & Genomics (NICOG) Centre, School of Psychology and Discipline of Biochemistry, National University of Ireland Galway, Galway H91 TK33, Ireland. [39]Medical Research Institute and Faculty of Health, University of Newcastle, New South Wa0les 2308, Australia. [40]Institute for Molecular Medicine Finland (FIMM), University of Helsinki, Helsinki FI-00014, Finland. [41]Centre for EpidemiologyDivision of Population Health, Health Services Research & Primary Care, The University of Manchester, Manchester M13 9PL, UK. [42]Department of Epidemiology and Biostatistics, School of Public Health, Imperial College London, London W2 1PG, UK. [43]Department of Psychology, University of California Riverside, Riverside 92521 CA, USA. [44]University of Bordeaux, Bordeaux Population Health Research Center, INSERM UMR 1219, F-33000 Bordeaux, France. [45]Institute for Medical Informatics, Statistics and Epidemiology, University of Leipzig, Leipzig 04107, Germany. [46]LIFE—Leipzig Research Center for Civilization Diseases, University of Leipzig, Leipzig 04107, Germany. [47]Department of Epidemiology, School of Public Health, University of Michigan, Ann Arbor, MI 48109, USA. [48]Survey Research Center, Institute for Social Research, University of Michigan, Ann Arbor, MI 48104, USA. [49]Icelandic Heart Association, Kopavogur IS-201, Iceland. [50]University of Iceland, Reykjavik 101, Iceland. [51]Department of Respiratory Medicine, Ghent University Hospital, De Pintelaan 185, 9000 Ghent, Belgium. [52]Section of Gerontology and Geriatrics, Department of Internal Medicine, Leiden University Medical Center, Leiden 2333, The Netherlands. [53]Department of Medicine, Division of Geriatrics, University of Mississippi Medical Center, Jackson 39216 MS, USA. [54]Queensland Brain Institute, University of Queensland, Brisbane 4072, Australia. [55]Centre for Advanced Imaging, University of Queensland, Brisbane 4072, Australia. [56]Rush Alzheimer's Disease Center, Rush University Medical Center, Chicago 60612 IL, USA. [57]Department of Neurological Sciences, Rush University Medical Center, Chicago 60612 IL, USA. [58]National Ageing Research Institute, Royal Melbourne Hospital, Victoria 3052, Australia. [59]Academic Unit for Psychiatry of Old Age,

University of Melbourne, St George's Hospital, Kew 3010, Australia. [60]Univ. Lille, Inserm, CHU Lille, Institut Pasteur de Lille, U1167-LabEx DISTALZ, F-59000 Lille, France. [61]Norwegian Centre for Mental Disorders Research (NORMENT), Institute of Clinical Medicine, University of Oslo, Oslo 0372, Norway. [62]Mathematics and Statistics, Murdoch University, Perth 6150, Australia. [63]Hunter Medical Research Institute and Faculty of Health, University of Newcastle, New South Wales 2305, Australia. [64]Department of NeurologyBryan Alzheimer's Disease Research Center, and Center for Genomic and Computational Biology, Duke University Medical Center, Durham 27708 NC, USA. [65]Psychiatry and Behavioral Sciences, Division of Medical Psychology, and Department of Neurology, Duke University Medical Center, Durham 27708 NC, USA. [66]Department of Psychiatry, Johns Hopkins University School of Medicine, MDBaltimore 21287, USA. [67]McKusick-Nathans Institute of Genetic Medicine, Johns Hopkins University School of Medicine, MD Baltimore 21287, USA. [68]Institute of Human Genetics, University of Bonn, Bonn 53113, Germany. [69]Department of Genomics, Life and Brain Center, University of Bonn, Bonn 53113, Germany. [70]Departments of Behavioral Sciences, Rush University Medical Center, Chicago 60612 IL, USA. [71]Dementia Centre for Research Collaboration, University of New South Wales, Sydney 2031 NSW, Australia. [72]Department of Psychology, Yale University, New Haven 06520 CT, USA. [73]Human Longevity Inc, Durham 27709 NC, USA. [74]UCLA Semel Institute for Neuroscience and Human Behavior, Los Angeles 90024 CA, USA. [75]23andMe, Inc., Mountain View 94041 CA, USA. [76]Department of Cognitive Science, University of California, San Diego, La Jolla 92093 CA, USA. [77]Department of Neurosciences, University of California, San Diego, La Jolla 92093 CA, USA. [78]Department of Radiology, University of California, San Diego, La Jolla 92093 CA, USA. [79]MRC-PHE Centre for Environment, School of Public Health, Imperial College London, London W2 1PG, UK. [80]Department of Psychology, Virginia Commonwealth University, Richmond 23284 VA, USA. [81]Clinical and Translational Neuroscience Branch, Intramural Research Program, National Institute of Mental Health, National Institute of Health, Bethesda 20892 MD, USA. [82]National Institute for Health and Welfare, Helsinki FI-00271, Finland. [83]Department of General Practice and Primary Health Care, University of Helsinki, Helsinki 00290, Finland. [84]Helsinki University Central Hospital, Unit of General Practice, Helsinki FI-00029, Finland. [85]Folkhälsan Research Centre, Helsinki 2018, Finland. [86]Robertson Centre for Biostatistics, University of Glasgow, Glasgow G12 8QQ, United Kingdom. [87]Department of Psychiatry, Martin Luther University of Halle-Wittenberg, Halle 06108, Germany. [88]Virginia Institute for Psychiatric and Behavioral Genetics, Virginia Commonwealth University, Richmond 23298 VA, USA. [89]QIMR Berghofer Medical Research Institute, Brisbane 4029, Australia. [90]Department of Neurology, Johns Hopkins University School of Medicine, Baltimore 21287 MD, USA. [91]Department of Epidemiology, Johns Hopkins Bloomberg School of Public Health, Baltimore 21205 MD, USA. [92]Department of Data Science, University of Mississippi Medical Center, Jackson 39216 MS, USA. [93]Intramural Research Program National Institutes on Aging, National Institutes of Health, Bethesda 20892 MD, USA. [94]Department of Psychiatry, National and Kapodistrian University of Athens Medical School, Eginition Hospital, Athens 11528, Greece. [95]University Mental Health Research Institute, Athens GR-156 01, Greece. [96]Neurobiology Research Institute, Theodor-Theohari Cozzika Foundation, Athens 11521, Greece. [97]Department of Epidemiology, University of North Carolina Gillings School of Global Public Health, Chapel Hill 27599 NC, USA. [98]Department of Clinical Physiology, Tampere University Hospital, and Finnish Cardiovascular Research Center, Tampere FI-33014, Finland. [99]Faculty of Medicine and Life Sciences, University of Tampere, Tampere 33521, Finland. [100]Department of Clinical Physiology, Finnish Cardiovascular Research Center—Tampere, Faculty of Medicine and Life Sciences, University of Tampere, Tampere 33014, Finland. [101]Department of Psychiatry Medical Faculty, University of Cologne, Cologne 50923, Germany. [102]Department of Psychiatry and Psychotherapy, University of Bonn, Bonn 53127, Germany. [103]German Center for Neurodegenerative Diseases (DZNE), Bonn 53127, Germany. [104]Department for Neurodegenerative Diseases and Geriatric Psychiatry, University of Bonn, Bonn 53127, Germany. [105]Department of Neurology, Mayo Clinic, Rochester 55905 MN, USA. [106]Neuropsychiatric Institute, Prince of Wales Hospital, Sydney 2031, Australia. [107]Brain and Mind Centre—The University of Sydney, Camperdown, NSW 2050, Australia. [108]School of Medical Sciences, University of New South Wales, Sydney 2052, Australia. [109]Department of Clinical Chemistry, Fimlab Laboratories, Finnish Cardiovascular Research Center—Tampere, Faculty of Medicine and Life Sciences, University of Tampere, Tampere 33014, Finland. [110]Department of Clinical Chemistry, Finnish Cardiovascular Research Center—Tampere, Faculty of Medicine and Life Sciences, University of Tampere, Tampere 33014, Finland. [111]Max Planck Institute for Human Development, Berlin 14195, Germany. [112]Technische Universität Dresden, Dresden 01187, Germany. [113]Genetic and Molecular Epidemiology Group, Lübeck Interdisciplinary Platform for Genome Analytics, Institutes of Neurogenetics & Cardiogenetics, University of Lübeck, Lübeck, Germany. [114]Department of Neurology, School of Medicine, University of Washington, Seattle 98195-6465 WA, USA. [115]Department of Epidemiology, University of Washington, Seattle 98195 WA, USA. [116]Department of Neurology and Psychiatry, University of Pittsburgh, Pittsburgh 15213 PA, USA. [117]Institute of Social Medicine, Occupational Health and Public Health (ISAP), University of Leipzig, Leipzig 04103, Germany. [118]Department of Biological and Medical Psychology, University of Bergen, Bergen 5009, Norway. [119]K. G. Jebsen Center for Neuropsychiatry, University of Bergen, Bergen N-5009, Norway. [120]Umeå Center for Functional Brain Imaging (UFBI), Umeå University, Umeå SE-901 87, Sweden. [121]Department of Statistics, USBE Umeå University, S-907 97 Umeå, Sweden. [122]Institute for Molecular Bioscience, University of Queensland, Brisbane 4072, Australia. [123]The Institute of Medical Sciences, Aberdeen Biomedical Imaging Centre, University of Aberdeen, Aberdeen AB25 2ZD, UK. [124]Division of Brain Sciences, Department of Medicine, Imperial College, London SW7 2AZ, UK. [125]Department of Radiation Sciences, Umeå University, Umeå SE-901 87, Sweden. [126]Department of Integrative Medical Biology, Umeå University, Umeå SE-901 87, Sweden. [127]Centre for Integrated Genomic Medical Research, Institute of Population Health, University of Manchester, Manchester M13 9PT, UK. [128]Karolinska Institutet, Aging Research Center, Stockholm University, Stockholm SE-113 30, Sweden. [129]Department of Psychology, School of Philosophy, Psychology and Language Sciences, The University of Edinburgh, Edinburgh EH8 9JZ, UK. [130]Gen-Info LLC, Zagreb 10000, Croatia. [131]Faculty of Medicine, University of Split, Split 21000, Croatia. [132]Department of Psychology, Stanford University, Palo Alto 94305-2130 CA, USA. [133]Deparment of Health Services, University of Washington, Seattle 98195-7660 WA, USA. [134]Kaiser Permanente Washington Health Research Institute, Seattle 98101 WA, USA. [135]Department of Developmental Disability Neuropsychiatry, School of Psychiatry, University of New South Wales, Sydney 2052, Australia. [136]Department of Psychological and Brain Sciences, Indiana University, Bloomington, IN 47405-7007, USA. [137]Institute for Translational Genomics and Population Sciences, Los Angeles Biomedical Research Institute at Harbor-UCLA Medical Center, Torrance, CA 90502, USA. [138]Department of Pediatrics, Harbor-UCLA Medical Center, Torrance 90509 CA, USA. [139]Department of Genetics and Genomic Science and Institute for Multiscale Biology, Icahn School of Medicine at Mount Sinai, New York 10029 NY, USA. [140]Mental Illness Research, Education, and Clinical Center (VISN 2), James J. Peters VA Medical Center, Bronx 10468 NY, USA. [141]Institute of Molecular Biology and Biochemistry, Centre for Molecular Medicine, Medical University of Graz, Graz 8036, Austria. [142]Robert and Beverly Lewis Center for Neuroimaging, University of Oregon, Eugene 97403 OR, USA. [143]Department of Neurology, Boston University School of Medicine, Boston 02118 MA, USA. [144]The National Heart, Lung, and Blood Institute's Framingham Heart Study, Framingham 01702-5827 MA, USA. [145]Department of Medical Biometry, Informatics and Epidemiology, University Hospital, Bonn D-53012, Germany. [146]Laboratory of NeuroGenetics, Department of Psychology & Neuroscience, Duke University, Durham 27708-0086 NC, USA. [147]Department of Molecular Epidemiology, Leiden University Medical Center, Leiden 2333, The Netherlands. [148]Institute of Cardiovascular and Medical Sciences, University of Glasgow, Glasgow G12 8QQ, United Kingdom. [149]Lieber Institute for Brain Development, Johns Hopkins University Medical Campus, Baltimore 21205 MD, USA. [150]Department of Hygiene and Epidemiology, University of Ioannina Medical School, Ioannina 45110, Greece. [151]Department of Public Health, University Hospital of Bordeaux, Bordeaux 33076, France. [152]Department of Internal Medicine, Erasmus Medical Center, Rotterdam 3015, The Netherlands. [153]Campbell

Family Mental Health Institute, Centre for Addiction and Mental Health, University of Toronto, Toronto M5T 1L8, Canada. [154]Department of Public Health, University of Helsinki, Helsinki 00014, Finland. [155]Department of Radiology, Erasmus MC, Rotterdam 3015, The Netherlands. [156]Department of Psychiatry and Behavioral Sciences, Faculty of Medicine, University of Crete, Heraklion GR-71003, Greece. [157]Human Genetics Center, School of Public Health, University of Texas Health Science Center at Houston, Houston 77030 TX, USA. [158]Human Genome Sequencing Center, Baylor College of Medicine, Houston 77030-3411 TX, USA. [159]Neuropsychiatric Genetics Research Group, Department of Psychiatry and Trinity College Institute of Neuroscience, Trinity College Dublin, Dublin DO2 AY89, Ireland. [160]Center for Translational and Systems Neuroimmunology, Department of Neurology, Columbia University Medical Center, New York 10032 NY, USA. [161]Program in Medical and Population Genetics, Broad Institute, Cambridge 02142 MA, USA. [162]Department of Neurology, University Hospital of Bordeaux, Bordeaux 33000, France. [163]Department of Global Health, University of Washington, Seattle 98104 WA, USA. [164]Department of Cardiology, Leiden University Medical Center, Leiden 2333, The Netherlands. [165]Institute for Behavioral Genetics, University of Colorado, Boulder 80309 CO, USA. [166]Wellcome Trust Sanger Institute, Wellcome Trust Genome Campus, Cambridge CB10 1SA, UK. [167]Department of Medical Genetics, University of Helsinki and University Central Hospital, Helsinki 00014, Finland. [168]Division of Neuroscience and Experimental Psychology, School of Biological Sciences, Manchester Academic Health Science Centre, and Manchester Medical School, Institute of Brain, Behaviour, and Mental Health, University of Manchester, Manchester M13 9PL, UK. [169]Department of Clinical Physiology and Nuclear Medicine, Turku University Hospital, Turku 20520, Finland. [170]School of Medicine and Public Health, University of Newcastle, New South Wales 2308, Australia. [171]Neuroscience Research Australia, Sydney 2031, Australia. [172]Faculty of Medicine, University of New South Wales, Sydney 2052, Australia. [173]Alzheimer Scotland Dementia Research Centre, University of Edinburgh, Edinburgh EH8 9JZ, UK. [174]Division of Nephrology and Hypertension, Department of Internal Medicine, Mayo Clinic, Rochester, MN 55905, USA. [175]Max Planck Institute for Human Cognitive and Brain Sciences, Leipzig 04103, Germany. [176]Day Clinic for Cognitive Neurology, University Hospital Leipzig, Leipzig 04103, Germany. [177]Division of Psychiatry Research, Zucker Hillside Hospital, Glen Oaks 11004 NY, USA. [178]Department of Psychiatry, Hofstra Northwell School of Medicine, Hempstead 11549 NY, USA. [179]Division of Psychiatry, University of Edinburgh, Edinburgh EH10 5HF, UK. [180]MRC Lifecourse Epidemiology Unit, University of Southampton, Southampton SO16 6YD, UK. [181]Glenn Biggs Institute for Alzheimer's and Neurodegenerative Diseases, University of Texas Health Sciences Center, San Antonio 78229 TX, USA. These authors contributed equally: Gail Davies, Max Lam. These authors jointly supervised this work: Todd Lencz, Ian J. Deary.

