## [peer review file · Nature Communications]

Reviewer #1 (Remarks to the Author):

Summary

The study combines results from new genetic analyses of cognitive function in UKB (full release) with summary statistics from cohort-level results from two previously published meta-analyses [conducted by the COGENT and CHARGE consortia, see (1, 2)]. Combining these three data sets results in a sample of 280K individuals. The paper identifies 99 independent loci at genome-wide significance (120 lead SNPs). The bulk of the paper is devoted to briefly summarizing the large number of analyses that were conducted on summary statistics from the 280K meta-analysis.

Major Comments

Overall, I think this paper has the potential to make a nice, incremental contribution to the literature on the genetics of cognitive function. It is clear that lots of effort went into the manuscript, but I also think the authors could do a lot to improve the paper's readability and clarify the significance of its findings. In particular, the authors need to do a better job explaining how the results relate to earlier work and those of other, contemporaneous, studies that address similar questions using mostly overlapping data. Many of the key results being highlighted sound like confirmation of findings already reported in previous studies based on smaller samples, but it is hard to tell for sure given the way the paper is currently written. I also think that in a number of cases, there is insufficient methodological detail provided. For one or two of the analyses, I really tried to get to the bottom of exactly what was done, and I was not persuaded that the analyses were always designed and executed as carefully as they could have been.

Most of the reported increase in the number of loci is accounted for by the addition of the large UKB sample, an incredible resource available to hundreds of researchers around the world. I can understand that the authors felt the urge to rush the paper off to publication given that many other groups surely have similar work in progress. For example:

1. A contemporaneous study, Savage et al. (3), also relies primarily on analyses of cognitive function in the full release of UKB (N = 280K). I will refer to it as the PGC paper below.
2. The senior author of this paper has a separate paper that reports results from a multivariate genetic analysis of largely overlapping data (4) and finds 107 loci associated with intelligence.
3. Finally, multiple groups have run GWASs of nearly all UKB phenotypes. At least one of these is making the summary statistics publicly available:

<https://sites.google.com/broadinstitute.org/ukbbgwasresults/home?authuser=0>

Thus, information about most of the 99 loci identified in this paper have probably been available online for a while to anyone interested - and ultimately it is the UKB resource that enabled their detection. I don't envy editors who have to make decisions about what constitutes a contribution in this new era of massive biobanks that everyone has access to.

Other Comments

1. The PGC paper is similar to this one in that it relies primarily on data from UKB (though supplemented by data from Sniekers et al. (5) instead of CHARGE and COGENT). The sample sizes of this study and the PGC paper are near-identical. Yet, the PGC study identifies 206 loci at genome-wide significance, whereas the current study reports 99. This large difference immediately seemed worrisome to me and I still don't understand what explains it. Here are some possibilities:

a. Was the genomic control applied at the cohort level in this study? If cohort-level GC was applied in UKB (and other large cohorts) then it would go some way toward explaining the smaller number of loci, despite the similar sample sizes. Cohort-level GC is extremely conservative when the discovery cohort has 200K people in it and the outcome is highly polygenic. Hence, applying cohort-level GC would yield very conservative SEs (and hence fewer loci). If this is the explanation, I am less alarmed since it simply means the authors of this study are reporting incorrect standard errors.

b. I tried understand how the list of 120 lead SNPs was generated and eventually decided the authors did some sort of LD-pruning with R2 cutoff of 0.1 and then merged SNPs that were less than 250 kb apart into a single locus. Is that correct? Is it possible that a more liberal R2 cutoff in the PGC study explains why the study finds >200 loci?

c. Inverse-variance weighted meta-analysis only makes sense if the coefficients are in the same units. Did the study have reasonable procedures in place to ensure that this was so? If so, what where they? If there are errors, inverse-variance weighting can lead to large efficiency losses (which could explain the paucity of loci relative to the other study). A simple sanity check is to do sample-size weighted meta-analysis instead and verify that the results are similar. [Since the UKB measure of cognitive function has the highest heritability, it is actually inefficient to do inverse-variance weighting – a weighting scheme incorporating the higher heritability in UKB would have been preferable – but there is not much that can be done about that at this point].

d. What is the average test statistic for SNPs with MAF>1% when you look separately at CHARGE, COGENT and UKB? What about the CHARGE/COGENT/UKB meta-analysis? If you plot the four averages on a graph with sample-size on the x-axis, is the relationship roughly linear? If the answer is no, I would again worry that is something is not right and this could explain in part why this study reports fewer loci than the PGC paper.

4. I believe there'd be a stronger case for publishing a cleaned-up version of this paper in a high-impact journal such as Nat Hum Behavior if its authors agreed to publicly release summary statistics from the CHARGE and COGENT meta-analyses in the event of publication (or at least the CHARGE/COGENT meta). As the authors note in the nicely crafted introduction of the paper, sample-size limitations remain the main barrier to progress in this area. The publication of this paper could be a great opportunity to make a tangible contribution toward solving the problem that the authors, along with others, have correctly identified. Do the authors plan to make summary statistics publicly available?

5. When I read the discussion of the results from the paper's various analyses I repeatedly had a number of reactions:

a. I want to know what your findings mean and what the analyses are doing conceptually, not the name of the software command you used.

b. I want to know what results are new and what results reinforce (or contradict) previously published claims.

6. In a few cases, the authors try to make comparisons to earlier studies, which is something I appreciate, though I am not yet persuaded the comparisons are done in a reasonable way. For example, the authors ask how many of the 99 loci are novel and come up with the answer 77. I have questions/comments about this conclusion:

a. It appears the paper defines a locus as novel if none of its lead SNPs (usually there is only one, but sometimes a locus has more than one) reached genome-wide significance in previously published studies of either intelligence (5) or educational attainment (6). Assuming I am right about the locus definition, I worry the paper's definition may overstate the number of associations that can plausibly be labelled novel. In Supplementary Table 14, the authors show that 15 out of 120 lead SNPs reached genome-wide significance in Okbay et al. (6). In some cases, however, the p-values are just above the p-value cutoff. Let me illustrate using rs32106494 as an example. Apparently it had $P = 8 \times 10^{-8}$ in Okbay et al. (6). Therefore, the locus is novel according to the paper's definition. But suppose the lead SNP is in strong LD ($R^2=0.95$) with a SNP that did reach genome-wide significance in Okbay et al.'s results. Would it really make sense to proclaim that a new locus has been found in that case? To say that a locus is novel is to imply a new genetic signal has been identified. To investigate the practical relevance of this concern about the locus definition, I took the 120 lead SNPs and used publicly available SSGAC summary statistics to check how many remained if we drop ones with $R^2 > 0.3$ with at least one SNP that reached genome-wide significance in Okbay et al. (6). My quick-and-dirty analysis used summary statistics available online and suggests that 30 of the lead SNPs end up getting dropped (compared to 15, if we use the authors' approach of just looking up the lead SNP). The number would surely increase further if I lowered the R^2 threshold and also used summary statistics from Sniekers et al. (5) to apply the same criterion. (I admit I was expecting a number even higher than 30.) What to do? At minimum, the authors could make it clearer what they did – ultimately any locus definition is arbitrary, so as long as the definition is clearly stated, I suppose it is in the eyes of the beholder to determine exactly what it means for a locus to be new. Ideally, however, I would like to see more careful analyses of the sort I sketched above.

b. Another reason I find the 77 number a bit misleading is that the comparison is made to previously published results, but published papers are not really an up-to-date reflection of the current state of knowledge, given publication lags and rapid progress. I think that at minimum, the authors may want to acknowledge contemporaneous efforts I cited above. Personally, I would be more interested in knowing how many new loci you identified relative to an analysis based on just UKB.

7. For FUMA, MAGMA, LDSC r_g , and many of the other analyses you report, I would like to know:

a. Which of the results are new compared to what previous applications of these (or similar) tools were applied in earlier published studies?

b. Which results reinforce earlier findings?

c. Which, if any, contradict earlier findings?

Right now, everything feels awfully formulaic. The data were fed into various tools, which produced results of various sorts. Those results are then reported, without any serious attempt to help readers interpret them in light of earlier work. As a reader, I found myself wondering what we have learnt that was not known from earlier work.

For example, I am not sure the GREML and LDSC results deserve so much space in the main text. Given that resources such as LDHUB (7) now exist and that Okbay, Sniekers and COGENT papers all reported LDSC estimates of r_g with a long laundry list of phenotypes, I don't see that these analyses are telling us anything new. At best, they are providing additional evidence for findings for which the existing evidence was already very strong. It also seems important to clarify how a "a novel genetic correlation" was defined. If the definition is just that the authors can reject a zero correlation at $P < 0.05$ whereas previous studies could not, I don't find that terribly interesting, unless there was also a meaningful change in the point estimate (especially if the magnitude of the r_g is really small, as appears to be the case for all new results except maybe the r_g with MDD). Right now, the paper could make a naive reader think that estimates of r_g with ADHD, depression, bipolar and longevity have never been reported before. For at least the first three phenotypes, I would be shocked if estimates weren't already on LDHub, a mouseclick away. If the contribution of this study was to shrink the standard error enough to rule out a zero correlation at $P < 0.05$, I don't see that as a major advance but an incremental contribution that can be summarized in the main text without taking up so much space.

References

1. J. W. Trampush et al., GWAS meta-analysis reveals novel loci and genetic correlates for general cognitive function: a report from the COGENT consortium. *Mol. Psychiatry*. 22, 336–345 (2017).
2. G. Davies et al., Genetic contributions to variation in general cognitive function: a meta-analysis of genome-wide association studies in the CHARGE consortium (N=53 949). *Mol. Psychiatry*. 20, 183–192 (2015).
3. J. E. Savage et al., GWAS meta-analysis (N=279,930) identifies new genes and functional links to intelligence. *bioRxiv* (2017).
4. W. D. Hill, G. Davies, A. M. McIntosh, C. R. Gale, I. J. Deary, A combined analysis of genetically correlated traits identifies 107 loci associated with intelligence. *bioRxiv* (2017).
5. S. Sniekers et al., Genome-wide association meta-analysis of 78,308 individuals identifies new loci and genes influencing human intelligence. *Nat Genet*. 49, 1107–1112 (2017).
6. A. Okbay et al., Genome-wide association study identifies 74 loci associated with educational attainment. *Nature*. 533, 539–542 (2016).
7. J. Zheng et al., LD Hub: a centralized database and web interface to perform LD score regression that maximizes the potential of summary level GWAS data for SNP heritability and genetic correlation analysis. *Bioinformatics*. 33, 272–279 (2017).

Reviewer #2 (Remarks to the Author):

This paper is based on 280K individuals and increases the number of loci for general cognitive function/intelligence from 18 in the Sniekers et al (2017) paper to 99 loci, when functional annotation is taken into account. It finds genetic correlation with a broad set of traits similar to Sniekers et al. The finding of gene set for neurogenesis and dentrite, which are similar again to categories in Sniekers et al. The addition of the GWAS for reaction time adds additional interest. The

methods contain too little detail to evaluate the results in places.

1. Abstract and/or results: The number of independent loci is the important metric. Please remove the total number of SNPs that are associated with $p < 5 \times 10^{-8}$ and the number of SNPs with suggestive associations. Also, please make clear in the abstract that the number of loci are based on analysis that takes annotation into account.

2. Please provide a clearer definition of general cognitive function at the start of the introduction.

3. 18 loci for intelligence were detected in Sniekers et al, which includes a substantial overlap (the UKBiobank) with this study. Why were only 4 loci from that study detected in this study? Please include a supplementary table with the same type of information as Table S14 that includes the 18 GWAS variants from Sniekers.

4. Multiple places in the results talk about completely novel results (defined those that were $p < .05$ in any previous studies). This emphasis doesn't make sense to me. The relevant new loci are those that have not been found before as genome-wide significant for cognition, regardless of the level of significance in previous studies. These parts of the results should be removed or expanded to include all new loci.

5. The gene-based tests in the manuscript are based on all variants within the gene body (though it is not clear from the methods exactly which test was performed within MAGMA). These are distinct from exome/transcript based tests which aim to point to specific genes based on potentially causal exomal variants. Gene-body tests have, at least, two serious issues:

1) they will miss genes for which causal regulatory variants occur outside the gene body/promotor region when there is weak LD between the gene-body SNPs and the causal variant;

2) for the strongest loci, many genes may appear to be significant because they contain SNPs relatively low LD with the lead SNP. For these genes there would be no evidence of association if the lead variant were accounted for in the analysis.

I would strongly advocate for their removal from the paper, or at a minimum, their removal from the abstract and much more cautious interpretation of what an associated gene means.

6. The Figure with the number of loci from previous studies does not add sufficient information over the Table 1 to be part of the paper. Please remove or change to a Supplemental Figure.

7. In the section on the lead and tagged SNPs within the 99 genetic loci, the functional characteristics of the SNPs present in the loci are described. Are there more SNPs with specific functional characteristics identified in these loci than would be expected by chance (taking into account the analysis performed to identify the loci)?

8. It is not clear how the different scales of the first PC and the fluid intelligence are harmonized, particularly as the data was combined using an inverse variance meta-analysis (which uses the Odds Ratios and standard errors). Were the general cognitive values inverse normalized before analysis, or, if not, how were they harmonized?

9. Please add the % of the variance explained by the first principal component account for in each study to the Supplementary Table 2. The methods state that the CHARGE study was checked for consistency of the first principle component. What does consistency mean, and how were the other studies checked?

10. In the supplementary table 2 please include the cutoff for relatedness for each cohort, if used. If no cutoff is used please describe how the analysis accounts for relatedness if it is not obvious from the program name or current methods description.

11. The use of FUMA will result in prioritization of variants that are annotated to enriched genomic features. The use of FUMA may cause loci to be missed if they don't have functional characteristics of other loci and also makes it harder to compare results across studies not using a similar strategy. Please also include Supplemental tables for general cognitive function and reaction time of lead SNPs for independent loci that does not use the genomic features to select the SNPs.

12. Is there evidence of heterogeneity effect sizes for each of the SNPs discovered across the individual component studies, for the lead SNPs for both the FUMA and the non-FUMA based SNPs. Please add the I² and p-value for heterozygosity to the tables for the FUMA and non-FUMA based results separately for CHARGE and COGENT meta-analysis, the UK Biobank meta-analysis and the CHARGE/COGENT & UKBIOBANK meta-analysis. Please add the OR and SE and p-value for the CHARGE/COGENT meta-analysis and separately for the UKBIOBANK studies in addition to the OR SE and p-value for the overall study.

13. The sentence on page 23, "Adjustments for age, sex and population stratification, if required, were included in the model" makes it sound like age and sex were adjusted for all analysis. In the Supplementary Table 2, there is column for additional covariates. The column has very few entries. Is this column intended to contain the adjustments in addition to age and sex, or to list age and sex if adjusted for? Very few studies list adjustment for PC's. Are these the only studies adjusting for PC's?

14. Please clarify text from page 123/124 of the Supplemental Material: SNPs in LD ($r^2 > .6$) with independent variants may not be in $r^2 > .6$ with lead SNPs. Please clarify what "in LD" means in the following sentence means, "The lead SNPs and those in LD with the lead SNPs are then mapped" ..

15. When referring to loci that overlap between cognitive function and traits in the GWAS catalogue, how is overlap defined? Does it mean the lead variant from this study is in high r^2 with the lead SNP for the GWAS catalogue trait?

Laura Scott

Reviewer #3 (Remarks to the Author):

The paper describes GWAS analyses on cognitive function in over 280,000 participants. It uses an established proxy measure for general cognitive function that is based on two independent

cognitive tests assessed in each cohort, thus enabling comparison of cohorts assessed with distinct cognitive measures. Using this approach it identifies 9714 SNPs of genome-wide significance, of which 120 were independent lead SNPs in 99 significant genomic loci. The authors thus have expertly generated an impressive dataset that is of immense potential value.

The paper then moves on to characterise these SNPs in a flurry of biometric analyses that may be informative, but do not provide much insight or develop hypotheses about the actual biological mechanisms that are identified by the genes discovered. This in my view is the main weakness of the paper.

For example, we learn through scores derived from CADD or Regulate DB that SNPs in the genetic loci might be deleterious or might affect function. However, there are no first hand data supporting and exactly quantifying these effects, nor is there a serious attempt (except for MAGMA gene set analyses providing an output at the most superficial level) to elucidate biochemical pathways and neural systems that may be affected by the functionality identified.

Thus the authors leave this extremely interesting dataset for others to interpret, and form and test hypotheses that can actually advance the field. This appears to be the explicit intention of the authors who describe their work as providing 'foundations for exploring mechanisms'. While there is merit in this approach there is also an element of dissatisfaction to this reviewer.

If the editors concur with the approach chosen by the authors I would not have any major comments about the expert execution of this work.

Reviewer comments

Reviewer 1

Summary

“The study combines results from new genetic analyses of cognitive function in UKB (full release) with summary statistics from cohort-level results from two previously published meta-analyses [conducted by the COGENT and CHARGE consortia, see (1, 2)]. Combining these three data sets results in a sample of 280K individuals. The paper identifies 99 independent loci at genome-wide significance (120 lead SNPs). The bulk of the paper is devoted to briefly summarizing the large number of analyses that were conducted on summary statistics from the 280K meta-analysis.”

Response: Although this is a descriptive rather than evaluative paragraph, we think a correction is required. We note that the CHARGE dataset presented by us in this paper is around double the size of the previously-published GWAS mentioned by the reviewer; for this study we have added ~52,000 new individuals.

Major Comments

“Overall, I think this paper has the potential to make a nice, incremental contribution to the literature on the genetics of cognitive function. It is clear that lots of effort went into the manuscript, but I also think the authors could do a lot to improve the paper’s readability and clarify the significance of its findings. In particular, the authors need to do a better job explaining how the results relate to earlier work and those of other, contemporaneous, studies that address similar questions using mostly overlapping data. Many of the key results being highlighted sound like confirmation of findings already reported in previous studies based on smaller samples, but it is hard to tell for sure given the way the paper is currently written. I also think that in a number of cases, there is insufficient methodological detail provided. For one or two of the analyses, I really tried to get to the bottom of exactly what was done, and I was not persuaded that the analyses were always designed and executed as carefully as they could have been.”

Response: We thank the reviewer for recognising the potential contribution of this manuscript to the literature.

We explain more clearly to the reader how the study has made many discoveries not found in other, contemporaneous, partly-overlapping studies. We now bring to prominence the novel findings of our study (58 novel loci), which we agree were not highlighted in the original manuscript (See p20). We highlight the 58 novel genomic loci associated with general cognitive function in Supplementary Table 16. We also highlight novel gene-based (291 novel genes) (See Supplementary Table 7) and gene-set associations (See p21).

We have gone through the descriptions of analyses in the paper. We agree with the referee that these must be clear enough to allow others to replicate them. We have now included more detailed explanations on the execution of the analyses in the methods section (p24-32).

“Most of the reported increase in the number of loci is accounted for by the addition of the large UKB sample, an incredible resource available to hundreds of researchers around the world. I can understand that the authors felt the urge to rush the paper off to publication given that many other groups surely have similar work in progress. For example:

1. A contemporaneous study, Savage et al. (3), also relies primarily on analyses of cognitive function in the full release of UKB (N = 280K). I will refer to it as the PGC paper below.

2. The senior author of this paper has a separate paper that reports results from a multivariate genetic analysis of largely overlapping data (4) and finds 107 loci associated with intelligence.

3. Finally, multiple groups have run GWASs of nearly all UKB phenotypes. At least one of these is making the summary statistics publicly available:

<https://sites.google.com/broadinstitute.org/ukbbgwasresults/home?authuser=0>

Thus, information about most of the 99 loci identified in this paper have probably been available online for a while to anyone interested - and ultimately it is the UKB resource that enabled their detection. I don't envy editors who have to make decisions about what constitutes a contribution in this new era of massive biobanks that everyone has access to."

Response: We agree that it is important to address this point about sample overlap in GWAS meta-analyses of cognitive function. We have done so in more detail in the revised paper. Our summary responses to this point are that the present report includes more than just the many new UK Biobank samples as its new resource/samples, and that we have found dozens of new genetic loci associated with cognitive function that the other studies mentioned above did not find.

In the present report we include additional CHARGE-associated samples that have not been analysed previously with UK Biobank, and are also not available online (14 new cohorts; ~52,000 samples). We don't agree that the loci we report in the present study are already available online; it is the collecting together of the many samples herein that has made this possible. Online resources that provide bare GWAS results aren't particularly interesting, informative or useful unless more in-depth/downstream analyses are undertaken and provided to complement and understand the results. In summary, then, neither the contemporaneous studies nor the online GWAS resources can provide the discoveries and interpretations which we offer in the present paper.

To make this clearer, in the revised paper we have included a table that details lookups of our 148 significant loci (an increase from the 99 in the original version because we added new samples) and compares them with Hill *et al.*, Okbay *et al.* and Sniekers *et al.* This identifies 58 novel loci (Supplementary Table 15) in our revised paper, which is a large novel contribution to the field. When compared to the Savage *et al.* results (this is what the referee refers to as the 'PGC' paper), we discover 41 novel loci. The Savage *et al.* results are not presented in the table of our revised paper, because these were obtained from a preprint manuscript on bioRxiv, and may not be the final results. However, we thought it was important to address this here because the paper was mentioned by the referee.

"1. The PGC paper is similar to this one in that it relies primarily on data from UKB (though supplemented by data from Sniekers *et al.* (5) instead of CHARGE and COGENT). The sample sizes of this study and the PGC paper are near-identical. Yet, the PGC study identifies 206 loci at genome-wide significance, whereas the current study reports 99. This large difference immediately seemed worrisome to me and I still don't understand what explains it. Here are some possibilities:"

Response: We have increased our sample size to ~300,000 and have now followed the suggestion of the reviewer to run an N-weighted meta-analysis. We now report 148 independent genome-wide significant loci. As stated above, there is not complete sample overlap between this study and the Savage *et al.* study. One major difference is the inclusion of a case-control sample in the Savage paper, with the 'cases' being people with very high IQs. The bioRxiv version of this manuscript states:

"Further, when excluding this sample from the meta-analysis, 78% of SNPs from the full meta-analysis retain 24 genome-wide significant (GWS; $P < 1 \times 10^{-8}$) association"

This statement suggests that, without this case-control sample, Savage *et al.* would be reporting around 161 significant loci. When compared to this result, our current reporting of 148 loci is similar. Valuably, though, the two papers do not report only the same loci.

We thank the reviewer for the suggestions below, which we have explored in detail and have used to improve our analyses. We provide responses to each of them.

“a. Was the genomic control applied at the cohort level in this study? If cohort-level GC was applied in UKB (and other large cohorts) then it would go some way toward explaining the smaller number of loci, despite the similar sample sizes. Cohort-level GC is extremely conservative when the discovery cohort has 200K people in it and the outcome is highly polygenic. Hence, applying cohort-level GC would yield very conservative SEs (and hence fewer loci). If this is the explanation, I am less alarmed since it simply means the authors of this study are reporting incorrect standard errors.”

Response: Cohort-level GC was applied to the original meta-analyses. We thank the reviewer for this comment and have now run the meta-analyses without GC. We used univariate LD score regression to check for evidence of stratification in the meta-analysis results. We report an LD score regression intercept close to 1 (1.058; SE = 0.011) and a ratio of 0.0659; this indicates that only 6.6% of the inflation observed can be ascribed to causes other than a polygenic signal. This analysis then returned an extra 26 loci, on top of our original 99. We then added some additional individuals which resulted in the 148 loci in total which we now report.

“b. I tried understand how the list of 120 lead SNPs was generated and eventually decided the authors did some sort of LD-pruning with R2 cutoff of 0.1 and then merged SNPs that were less than 250 kb apart into a single locus. Is that correct? Is it possible that a more liberal R2 cutoff in the PGC study explains why the study finds >200 loci?”

Response: We are grateful for the opportunity to make this clearer. FUMA was used to generate a list of 178 lead SNPs. The 178 lead SNPs were a subset of the 434 independent significant SNPs and were defined as independent significant SNPs that were independent from each other at $R^2 < 0.1$. Loci that were 250kb or closer, were merged into a single locus to create 148 risk loci. This is the same method used by Savage *et al.* (the ‘PGC’ study).

“c. Inverse-variance weighted meta-analysis only makes sense if the coefficients are in the same units. Did the study have reasonable procedures in place to ensure that this was so? If so, what where they? If there are errors, inverse-variance weighting can lead to large efficiency losses (which could explain the paucity of loci relative to the other study). A simple sanity check is to do sample-size weighted meta-analysis instead and verify that the results are similar. [Since the UKB measure of cognitive function has the highest heritability, it is actually inefficient to do inverse-variance weighting – a weighting scheme incorporating the higher heritability in UKB would have been preferable – but there is not much that can be done about that at this point].”

Response: We agree that inverse-variance weighted meta-analysis was the incorrect method to use. We have now re-run the meta-analysis using the N-weighted method.

“d. What is the average test statistic for SNPs with MAF>1% when you look separately at CHARGE, COGENT and UKB? What about the CHARGE/COGENT/UKB meta-analysis? If you plot the four averages on a graph with sample-size on the x-axis, is the relationship roughly linear? If the answer is no, I would again worry that is something is not right and this could explain in part why this study reports fewer loci than the PGC paper.”

Response: We are grateful for this suggestion. We have plotted these values, as requested, and include the graph below. From the graph we observe that the relationship is roughly linear ($r^2 = 0.896$).

“4. I believe there'd be a stronger case for publishing a cleaned-up version of this paper in a high-impact journal such as Nat Hum Behavior if its authors agreed to publicly release summary statistics from the CHARGE and COGENT meta-analyses in the event of publication (or at least the CHARGE/COGENT meta). As the authors note in the nicely crafted introduction of the paper, sample-size limitations remain the main barrier to progress in this area. The publication of this paper could be a great opportunity to make a tangible contribution toward solving the problem that the authors, along with others, have correctly identified. Do the authors plan to make summary statistics publicly available?”

Response: We are grateful for the positive comment about the Introduction. The UK Biobank results will be made available online, as we have done previously with the UK Biobank's partial-release of genetic data a few years ago. The CHARGE and COGENT results will be available by application to the respective consortium; we do not lead these consortia, which are led by Dr Jan Bressler and Dr Todd Lencz, respectively.

“5. When I read the discussion of the results from the paper's various analyses I repeatedly had a number of reactions:

a. I want to know what your findings mean and what the analyses are doing conceptually, not the name of the software command you used.

Response: We agree with this comment, and we have now revised all relevant sections, and we now describe the method to make the ideas behind the analyses the primary concern. See p26-32.

b. I want to know what results are new and what results reinforce (or contradict) previously published claims. “

Response: As mentioned above, we now highlight the 58 novel SNP-based findings in the main text and in Supplementary Table 16. Also, the replicated findings from previous studies (Hill *et al.*, Sniekers *et al.* and Okbay *et al.*) are highlighted in Supplementary Table 15. The 291 novel gene-based findings are highlighted in the main text and in Supplementary Table 7.

The following has been added to the main text (p 15).

“A comparison of these 148 loci with results from the largest previous GWASs of cognitive function¹⁶, and educational attainment²⁴, and an MTAG analysis of cognitive function¹⁷—which included a subsample of individuals contributing to the present study—confirmed that 11, 24, and 89 of these were, respectively, genome-wide significant in these previous studies (Supplementary Table 15). Of the 148 loci found in the present study, 58 have not been reported previously in other GWA studies of cognitive function or educational attainment (novel loci are indicated in Supplementary Table 16).”

“These 709 genes were compared to gene-based associations from previous studies of general cognitive function and educational attainment^{15-17,25}; 418 were replicated in the present study, and 291 were novel. The 291 new gene-based associations are highlighted in Supplementary Table 7.”

“6. In a few cases, the authors try to make comparisons to earlier studies, which is something I appreciate, though I am not yet persuaded the comparisons are done in a reasonable way. For example, the authors ask how many of the 99 loci are novel and come up with the answer 77. I have questions/comments about this conclusion:

a. It appears the paper defines a locus as novel if none of its lead SNPs (usually there is only one, but sometimes a locus has more than one) reached genome-wide significance in previously published studies of either intelligence (5) or educational attainment (6). Assuming I am right about the locus definition, I worry the paper’s definition may overstate the number of associations that can plausibly be labelled novel. In Supplementary Table 14, the authors show that 15 out of 120 lead SNPs reached genome-wide significance in Okbay et al. (6). In some cases, however, the p-values are just above the p-value cutoff. Let me illustrate using rs32106494 as an example. Apparently it had $P = 8 \times 10^{-8}$ in Okbay et al. (6). Therefore, the locus is novel according to the paper’s definition. But suppose the lead SNP is in strong LD ($R^2=0.95$) with a SNP that did reach genome-wide significance in Okbay et al.’s results. Would it really make sense to proclaim that a new locus has been found in that case? To say that a locus is novel is to imply a new genetic signal has been identified. To investigate the practical relevance of this concern about the locus definition, I took the 120 lead SNPs and used publicly available SSGAC summary statistics to check how many remained if we drop ones with $R^2 > 0.3$ with at least one SNP that reached genome-wide significance in Okbay et al. (6). My quick-and-dirty analysis used summary statistics available online and suggests that 30 of the lead SNPs end up getting dropped (compared to 15, if we use the authors’ approach of just looking up the lead SNP). The number would surely increase further if I lowered the R^2 threshold and also used summary statistics from Sniekers et al. (5) to apply the same criterion. (I admit I was expecting a number even higher than 30.) What to do? At minimum, the authors could make it clearer what they did – ultimately any locus definition is arbitrary, so as long as the definition is clearly stated, I suppose it is in the eyes of the beholder to determine exactly what it means for a locus to be new. Ideally, however, I would like to see more careful analyses of the sort I sketched above.

Response: We are grateful for this comment, we agree with it, and we have done the new work needed to address it, and to make sure that loci we are reporting as novel are indeed credibly novel. We now more carefully assess the significant loci and perform look-ups on all tagged SNPs ($r^2 > 0.6$) within each locus, including all 1000 genomes SNPs. We now class any tagged SNP previously reported as genome-wide significant, as replication. These look-ups, which we did for the results reported by Hill et al., Sniekers et al. and Okbay et al., are presented in Supplementary Table 15. Of the 148 significant loci, 58 are novel based on this updated look-up protocol.

b. Another reason I find the 77 number a bit misleading is that the comparison is made to previously published results, but published papers are not really an up-to-date reflection of the current state of

knowledge, given publication lags and rapid progress. I think that at minimum, the authors may want to acknowledge contemporaneous efforts I cited above. Personally, I would be more interested in knowing how many new loci you identified relative to an analysis based on just UKB.”

Response: Again, we are grateful for this suggestion and we have done the work needed to answer it. Ninety-two new loci are identified in the meta-analysis of CHARGE, COGENT and UK Biobank, when compared to an analysis based only on UK Biobank. Of these 92, 51 are novel findings. We now include this number of novel loci that become significant when CHARGE and COGENT are added to UK Biobank in the main text. See p23.

Inserted text: “Both the overall size of the present study’s meta-analysis of GWASs and the inclusion of a single large sample, UK Biobank, are strengths, which contributed to the abundance of new findings. When compared to an analysis of only UK Biobank herein, the current meta-analysis adds 92 independent significant loci, 51 of which are novel.”

We have compared our results to the Savage *et al.* paper (the ‘PGC’ paper) and we find 41 loci to be novel. We have not included this in the main text of the paper, as at present the Savage paper is not published and the preprint version (on bioRxiv) may not present the final published results.

“7. For FUMA, MAGMA, LDSC rg, and many of the other analyses you report, I would like to know:

- a. Which of the results are new compared to what previous applications of these (or similar) tools were applied in earlier published studies?
- b. Which results reinforce earlier findings?
- c. Which, if any, contradict earlier findings? “

Response: We agree that these points should be made clearer. We now highlight novel findings in the text and in the corresponding supplementary table. Novel genomic loci for general cognitive function are highlighted in Supplementary Table 16. Novel gene-based findings for general cognitive function are highlighted in Supplementary Table 7. Novel gene-sets are highlighted in the text (See p21).

Details of all replicated findings have also been added to the discussion. See p19-22.

“Right now, everything feels awfully formulaic. The data were fed into various tools, which produced results of various sorts. Those results are then reported, without any serious attempt to help readers interpret them in light of earlier work. As a reader, I found myself wondering what we have learnt that was not known from earlier work.”

Response: We agree that the original manuscript may have appeared somewhat formulaic and did not provide enough insight or develop hypotheses about the actual biological mechanisms that are identified by the genes discovered; this was in part because of the restrictions of the concise ‘Letter’ format, which is no longer necessary. We have re-written sections of the paper to focus on the novel findings and to bring the information together more cohesively in the context of existing knowledge, with an emphasis on their value. See p15, 17

Inserted text: “A comparison of these 148 loci with results from the largest previous GWASs of cognitive function¹⁶, and educational attainment²⁴, and an MTAG analysis of cognitive function¹⁷—which included a subsample of individuals contributing to the present study—confirmed that 11, 24, and 89 of these were, respectively, genome-wide significant in these previous studies (Supplementary Table 15). Of the 148 loci found in the

present study, 58 have not been reported previously in other GWA studies of cognitive function or educational attainment (novel loci are indicated in Supplementary Table 16)."

"These 709 genes were compared to gene-based associations from previous studies of general cognitive function and educational attainment^{13,16,17,25}; 418 were replicated in the present study, and 291 were novel. The 291 new gene-based associations are highlighted in Supplementary Table 7."

"We report, for the first time, significant genetic correlations between general cognitive function and: hypertension ($r_g = -0.15$, $SE = 0.02$), grip strength (right hand: $r_g = 0.09$, $SE = 0.02$), wearing glasses or contact lenses ($r_g = 0.28$, $SE = 0.04$), short-sightedness ($r_g = 0.32$, $SE = 0.03$), long-sightedness ($r_g = -0.21$, $SE = 0.05$), heart attack ($r_g = -0.17$, $SE = 0.03$), angina ($r_g = -0.18$, $SE = 0.03$), lung cancer ($r_g = -0.26$, $SE = 0.05$), and osteoarthritis ($r_g = -0.24$, $SE = 0.04$). We also report a significant genetic correlation with major depressive disorder ($r_g = -0.30$, $SE = 0.04$); this result strengthens previously-reported non-significant correlations of around -0.10 ^{16,17}."

"For example, I am not sure the GREML and LDSC results deserve so much space in the main text. Given that resources such as LDHUB (7) now exist and that Okbay, Sniekers and COGENT papers all reported LDSC estimates of r_g with a long laundry list of phenotypes, I don't see that these analyses are telling us anything new. At best, they are providing additional evidence for findings for which the existing evidence was already very strong. It also seems important to clarify how a "a novel genetic correlation" was defined. If the definition is just that the authors can reject a zero correlation at $P < 0.05$ whereas previous studies could not, I don't find that terribly interesting, unless there was also a meaningful change in the point estimate (especially if the magnitude of the r_g is really small, as appears to be the case for all new results except maybe the r_g with MDD). Right now, the paper could make a naive reader think that estimates of r_g with ADHD, depression, bipolar and longevity have never been reported before. For at least the first three phenotypes, I would be shocked if estimates weren't already on LDHub, a mouseclick away. If the contribution of this study was to shrink the standard error enough to rule out a zero correlation at $P < 0.05$, I don't see that as a major advance but an incremental contribution that can be summarized in the main text without taking up so much space."

Response: We agree with the reviewer and have reduced the space assigned to presenting these findings. See p16-17.

Revised text: "We estimated the proportion of variance explained by all common SNPs using GCTA-GREML in four of the largest individual samples: English Longitudinal Study of Ageing (ELSA: $N = 6,661$, $h^2 = 0.12$, $SE = 0.06$), Understanding Society ($N = 7,841$, $h^2 = 0.17$, $SE = 0.04$), UK Biobank Assessment Centre ($N = 86,010$, $h^2 = 0.25$, $SE = 0.006$), and Generation Scotland ($N = 6,507$, $h^2 = 0.20$, $SE = 0.05$ ²³) (Table 2). Genetic correlations for general cognitive function amongst these cohorts, estimated using bivariate GCTA-GREML, ranged from $r_g = 0.88$ to 1.0 (Table 2). These results indicate that the same genetic variants contribute towards phenotypic differences in general cognitive function across each of these three samples."

"We tested the genetic correlations between general cognitive function and 52 health-related traits. Thirty-six of these health-related traits were significantly genetically correlated with general cognitive function (Supplementary Table 12). We report, for the first time, significant genetic correlations between general cognitive function and: hypertension ($r_g = -0.15$, $SE = 0.02$), grip strength (right hand: $r_g = 0.09$, $SE = 0.02$), wearing glasses or contact lenses ($r_g = 0.28$, $SE = 0.04$), short-sightedness ($r_g = 0.32$, $SE = 0.03$), long-

sightedness ($r_g = -0.21$, $SE = 0.05$), heart attack ($r_g = -0.17$, $SE = 0.03$), angina ($r_g = -0.18$, $SE = 0.03$), lung cancer ($r_g = -0.26$, $SE = 0.05$), and osteoarthritis ($r_g = -0.24$, $SE = 0.04$). We also report a significant genetic correlation with major depressive disorder ($r_g = -0.30$, $SE = 0.04$); this result strengthens previously-reported non-significant correlations of around -0.10 ^{16,17}. We also note the important genetic association between general cognitive function and longevity ($r_g = 0.17$, $SE = 0.06$)."

Reviewer 2

Summary

"This paper is based on 280K individuals and increases the number of loci for general cognitive function/intelligence from 18 in the Sniekers et al (2017) paper to 99 loci, when functional annotation is taken into account. It finds genetic correlation with a broad set of traits similar to Sniekers et al. The finding of gene set for neurogenesis and dendrite, which are similar again to categories in Sniekers et al. The addition of the GWAS for reaction time adds additional interest. The methods contain too little detail to evaluate the results in places."

Response: We are grateful for the opportunity to make these changes. The revised paper now has more individuals, as described above. We now make it clear what is novel when compared with previous papers. There is more detail on methods and results. These are explained in detail below.

We note that only one gene set, "regulation of cell development", was significant in Sniekers *et al*. The gene sets involving neurogenesis and dendrite although described in detail by Sniekers *et al.*, were not significant findings in that study.

Major comments

"1. Abstract and/or results: The number of independent loci is the important metric. Please remove the total number of SNPs that are associated with $p < 5 \times 10^{-8}$ and the number of SNPs with suggestive associations. Also, please make clear in the abstract that the number of loci are based on analysis that takes annotation into account. "

Response: We have removed the reference to SNP number in the abstract, as suggested. The numbers of independent loci were not based on an analysis that takes annotation into account. More detail has been added to the methods to make clearer how FUMA was used to define these loci, as can be seen in the revised text, copied from the revised paper, below. See p26.

Revised text: "*Genomic Risk Loci (FUMA). Genomic risk loci were defined from the SNP-based association results, using FUnctional Mapping and Annotation of genetic associations (FUMA)*²³. *Firstly, independent significant SNPs were identified using the SNP2GENE function and defined as SNPs with a P-value of $\leq 5 \times 10^{-8}$ and independent of other genome wide significant SNPs at $r^2 < 0.6$. Using these independent significant SNPs, candidate SNPs to be used in subsequent annotations were identified as all SNPs that had a MAF ≥ 0.0005 and were in LD of $r^2 \geq 0.6$ with at least one of the independent significant SNPs. These candidate SNPs included those from the 1000 genomes reference panel and need not have been included in the GWAS performed in the current study. Lead SNPs were also identified using the independent significant SNPs and were defined as those that were independent from each other at $r^2 < 0.1$. Genomic risk loci that were 250kb or closer were merged into a single locus.*"

"2. Please provide a clearer definition of general cognitive function at the start of the introduction. "

Response: We agree that this is a useful change to make. This has been done. See p12-14.

“3. 18 loci for intelligence were detected in Sniekers et al, which includes a substantial overlap (the UKBiobank) with this study. Why were only 4 loci from that study detected in this study? Please include a supplementary table with the same type of information as Table S14 that includes the 18 GWAS variants from Sniekers. “

Response: We are grateful for this suggestion. In the revised paper we have provided more extensive information so that readers are clear about the new findings discovered in the present study. We performed a look-up of all tagged SNPs ($r^2 > 0.6$) within our 148 significant loci in the Sniekers summary results; these are presented in Table 15. The number of replicated loci, 11, has been updated in the text. See p15.

Text inserted in the revised version: “A comparison of these 148 loci with results from the largest previous GWASs of cognitive function¹⁶, and educational attainment²⁴, and an MTAG analysis of cognitive function¹⁷—which included a subsample of individuals contributing to the present study—confirmed that 11, 24, and 89 of these were, respectively, genome-wide significant in these previous studies (Supplementary Table 15). Of the 148 loci found in the present study, 58 have not been reported previously in other GWA studies of cognitive function or educational attainment (novel loci are indicated in Supplementary Table 16).”

“4. Multiple places in the results talk about completely novel results (defined those that were $p < .05$ in any previous studies). This emphasis doesn’t make sense to me. The relevant new loci are those that have not been found before as genome-wide significant for cognition, regardless of the level of significance in previous studies. These parts of the results should be removed or expanded to include all new loci. “

Response: We agree, and we take the reviewer’s advice and now consider any locus that did not previously attain genome-wide significance as novel. In addition, we think it is also relevant to note previous significant SNPs in LD ($r^2 > 0.6$) with variants in our locus, which are now accepted as replication in line with reviewer 1’s comment. See p27.

Inserted text: “Identifying novel findings. Previous evidence of association for each of the 148 genetic loci identified herein as being associated with general cognitive function was sought in the largest published GWASs of general cognitive function^{16,17} and education²⁴. We performed look-ups on all tagged SNPs ($r^2 > 0.6$) within each locus, including all 1000 genomes SNPs, and classed any tagged SNP previously reported as genome-wide significant, as replication. Details of these findings are presented in Supplementary Table 15.”

“5. The gene-based tests in the manuscript are based on all variants within the gene body (though it is not clear from the methods exactly which test was performed within MAGMA). These are distinct from exome/transcript based tests which aim to point to specific genes based on potentially causal exomal variants. Gene-body tests have, at least, two serious issues:

- 1) they will miss genes for which causal regulatory variants occur outside the gene body/promotor region when there is weak LD between the gene-body SNPs and the causal variant;
- 2) for the strongest loci, many genes may appear to be significant because they contain SNPs relatively low LD with the lead SNP. For these genes there would be no evidence of association if the lead variant were accounted for in the analysis.

I would strongly advocate for their removal from the paper, or at a minimum, their removal from the abstract and much more cautious interpretation of what an associated gene means. “

Response: We are grateful for this helpful comment, and we agree with the reviewer that the assumptions made when implementing a gene-based analysis need to be clearly presented within the manuscript. Below, we respond to the comments of the referee in detail, and then show the revised text we have written.

1). The reviewer is correct that “gene-body tests” will not detect an association if the SNPs within the body of the gene are in poor LD with a causal variant outside the gene-body. However, what gene-body tests can do is to detect the association from across the body of the gene where there are small, but consistent, associations with the phenotype being investigated. Gene-body tests conducted in this fashion have been shown to result in an increase in statistical power compared to single SNP analysis because they reduce the multiple testing burden, and the sum of the signal across many SNPs within the body of a gene is greater.^{1,2}

2). The gene-based analysis conducted in MAGMA utilises all SNPs in the data set that fall within the boundary of a gene, and as such there was no lead SNP in the MAGMA analysis. Rather, the definition of independent loci, which will contain at least one lead SNP, is a separate analysis from the gene-based statistics derived using MAGMA. The test carried out using MAGMA, as implemented in FUMA, was the default SNP-wise test using the mean χ^2 statistic derived on a per gene basis. The test used is more sensitive to association from across the gene, and differs from the MAGMA SNP-wise test using the top χ^2 statistic within a gene, which was designed to be more sensitive in instances where a small proportion of the gene is associated with the phenotype. In addition, linkage disequilibrium within and between genes, gene density, and gene length, are controlled for in the MAGMA analysis, ensuring that these properties of a gene do not contribute towards erroneous associations.³

We would respectfully disagree with reviewer 2 regarding the removal of the gene-based statistics. The assumption of gene-based analysis is that the SNPs within genes are of particular importance to the phenotype, in this case general cognitive function. A recent meta-analysis of education and general cognitive function has shown that SNPs located within protein coding regions are enriched for heritability of general cognitive function.⁴ In addition, further analyses aimed at elucidating the biological functions that can be perturbed by common genetic variance (gene-set analysis, gene property analysis, etc.) is simplified when one is able to use the gene as the statistical unit of association.

We have now included the text below describing the strengths and limitations of gene-based testing, on page 23.

Inserted text: “Gene-based analysis has been shown to increase the power to detect associations, because the multiple testing burden is reduced, and the effects of multiple SNPs are combined together. However, we note that gene-based testing will not be able to detect associations that fall outside of the gene-body. This means that, if SNPs in promoter regions harbour variants that are causal to differences in general cognitive function or reaction time, they will be missed in our gene-based analyses. In terms of localising more proximal structural and functional causes of variation in cognitive function, researchers could prioritise the genetic loci uncovered here that overlap with brain-related measures.”

1. Hill, W.D. et al. Functional Gene Group Analysis Indicates No Role for Heterotrimeric G Proteins in Cognitive Ability. PLoS one 9(2014).

2. Liu, J.Z. et al. A versatile gene-based test for genome-wide association studies. *Am. J. Hum. Genet.* 87, 139-45 (2010).
3. de Leeuw, C.A., Mooij, J.M., Heskes, T. & Posthuma, D. MAGMA: Generalized Gene-Set Analysis of GWAS Data. *PLoS Computational Biology* 11(2015).
4. Hill, W.D. et al. A combined analysis of genetically correlated traits identifies 187 loci and a role for neurogenesis and myelination in intelligence. *Molecular Psychiatry* (2018).

"6. The Figure with the number of loci from previous studies does not add sufficient information over the Table 1 to be part of the paper. Please remove or change to a Supplemental Figure. "

Response: As requested, we have moved the figure to Supplementary Information.

"7. In the section on the lead and tagged SNPs within the 99 genetic loci, the functional characteristics of the SNPs present in the loci are described. Are there more SNPs with specific functional characteristics identified in these loci than would be expected by chance (taking into account the analysis performed to identify the loci)?"

Response: We have removed the numbers of SNPs from this section; full details are still presented in Supplementary Table 16. These numbers of functional characteristics were presented only as descriptive information for the loci in the original manuscript. We were not able to identify a method to create an appropriate null distribution to allow us to test if there are more SNPs with specific functional characteristics identified than would be expected by chance. As described above, the method used to identify the loci did not take into account functional annotation. See p15.

Revised text: "Across many of the loci there is clear evidence of functionality including involvement in gene regulation, deleterious SNPs, eQTLs, and regions of open chromatin."

"8. It is not clear how the different scales of the first PC and the fluid intelligence are harmonized, particularly as the data was combined using an inverse variance meta-analysis (which uses the Odd Ratios and standard errors). Were the general cognitive values inverse normalized before analysis, or, if not, how were they harmonized? "

Response: We thank the reviewer for highlighting this. We had incorrectly applied inverse variance meta-analysis, and have now re-run the analysis with an N-weighted method.

"9. Please add the % of the variance explained by the first principal component account for in each study to the Supplementary Table 2. The methods state that the CHARGE study was checked for consistency of the first principle component. What does consistency mean, and how were the other studies checked? "

Response: We have now added the % of the variance explained by the first principal component account for each study to Supplementary Table 2.

Regarding the meaning of consistency, we use it to refer to "a consistent method of extracting a general cognitive function component". We have clarified the text further:
Revised text: "Principal component analyses results for the CHARGE cohorts were checked by one author (IJD) to establish the presence of a single component. The scree slope was examined, the percentage of variance accounted for by the first unrotated principal component was noted, and it was checked that all tests had sufficient loading on the first unrotated principal component. Scores on the first unrotated component were used as the cognitive phenotype (general cognitive function)."

“10. In the supplementary table 2 please include the cutoff for relatedness for each cohort, if used. If no cutoff is used please describe how the analysis accounts for relatedness if it is not obvious from the program name or current methods description. “

Response: No relatedness cut-offs were specified in the analysis protocol. The majority of cohorts in this study are population-based and consist of largely unrelated individuals. However, all cohorts were advised to fit any cohort-specific adjustments, which include accounting for relatedness, in their analysis. These cohort-specific adjustments are indicated in the ‘Additional covariates’ column in Supplementary Table 2.

“11. The use of FUMA will result in prioritization of variants that are annotated to enriched genomic features. The use of FUMA may cause loci to be missed if they don’t have functional characteristics of other loci and also makes it harder to compare results across studies not using a similar strategy. Please also include Supplemental tables for general cognitive function and reaction time of lead SNPs for independent loci that does not use the genomic features to select the SNPs. “

Response: We did not use the gene prioritization function of FUMA. We used FUMA to identify independent significant SNPs, lead SNPs, and genomic risk loci (which is very similar to clumping used by PLINK). At no point were functional annotations used to define significant SNPs or loci. We realise that this may not have been clear in the original manuscript and we thank the reviewer for highlighting this. We have added more detail to clarify this process in our methods. See p26.

Revised text: “Genomic Risk Loci (FUMA). Genomic risk loci were defined from the SNP-based association results, using FUnctional Mapping and Annotation of genetic associations (FUMA)²³. Firstly, independent significant SNPs were identified using the SNP2GENE function and defined as SNPs with a P-value of $\leq 5 \times 10^{-8}$ and independent of other genome wide significant SNPs at $r^2 < 0.6$. Using these independent significant SNPs, candidate SNPs to be used in subsequent annotations were identified as all SNPs that had a MAF ≥ 0.0005 and were in LD of $r^2 \geq 0.6$ with at least one of the independent significant SNPs. These candidate SNPs included those from the 1000 genomes reference panel and need not have been included in the GWAS performed in the current study. Lead SNPs were also identified using the independent significant SNPs and were defined as those that were independent from each other at $r^2 < 0.1$. Genomic risk loci that were 250kb or closer were merged into a single locus.”

“12. Is there evidence of heterogeneity effect sizes for each of the SNPs discovered across the individual component studies, for the lead SNPs for both the FUMA and the non-FUMA based SNPs. Please add the I2 and p-value for heterozygosity to the tables for the FUMA and non-FUMA based results separately for CHARGE and COGENT meta-analysis, the UK Biobank meta-analysis and the CHARGE/COGENT & UKBIOBANK meta-analysis. Please add the OR and SE and p-value for the CHARGE/COGENT meta-analysis and separately for the UKBIOBANK studies in addition to the OR SE and p-value for the overall study.”

Response: There is evidence of moderate heterogeneity for a small number of the lead SNPs (<2%). We have now added the requested heterogeneity information for the meta-analysis to Supplementary Table 3. We have also added Z-scores, p-values and heterogeneity information for separate meta-analyses of CHARGE, COGENT and UK Biobank.

“13. The sentence on page 23, “Adjustments for age, sex and population stratification, if required, were included in the model” makes it sound like age and sex were adjusted for all analysis. In the Supplementary Table 2, there is column for additional covariates. The column has very few entries. Is this column intended to contain the adjustments in addition to age and sex, or to list age and sex if

adjusted for? Very few studies list adjustment for PC's. Are these the only studies adjusting for PC's?"

Response: We have now made this sentence clearer in the text. We apologise for not keeping this consistent. The column referred to now refers to the additional covariates that were adjusted for. Age and sex were standard in all models. See p26.

Revised text: *"Adjustments for age, sex, and population stratification were included in the model for each cohort. Cohort-specific covariates—for example, site or familial relationships—were also fitted as required. Cohort-specific quality control procedures, imputation methods, and covariates are described in Supplementary Table 2."*

"14. Please clarify text from page 123/124 of the Supplemental Material: SNPs in LD ($r^2 > .6$) with independent variants may not be in $r^2 > .6$ with lead SNPs. Please clarify what "in LD" means in the following sentence means, "The lead SNPs and those in LD with the lead SNPs are then mapped" .."

Response: We thank the reviewer for highlighting this inconsistency in terminology. We have corrected this in the text.

Tagged SNPs used in subsequent annotations were identified as all SNPs that had a MAF ≥ 0.0005 and were in LD of $r^2 \geq 0.6$ with at least one of the independent significant SNPs. In our case there were 434 independent significant SNPs and 22,377 tagged SNPs. See Supplementary Table 16. We have revised the text to make this clearer. See p26 and 31.

Revised text: *"Using these independent significant SNPs, tagged SNPs to be used in subsequent annotations were identified as all SNPs that had a MAF ≥ 0.0005 and were in LD of $r^2 \geq 0.6$ with at least one of the independent significant SNPs. These tagged SNPs included those from the 1000 genomes reference panel and need not have been included in the GWAS performed in the current study."*

"Functional Annotation (implemented in FUMA)²³. The independent significant SNPs and those in LD with the independent significant SNPs were annotated for functional consequences on gene functions using ANNOVAR⁷³ and the Ensembl genes build 85. A CADD score⁷⁴, RegulomeDB score⁷⁵, and 15-core chromatin states⁷⁶⁻⁷⁸ were obtained for each SNP. eQTL information was obtained from the following databases: GTEx (<http://www.gtexportal.org/home/>), BRAINEAC (<http://www.braineac.org/>), Blood eQTL Browser (<http://genenetwork.nl/bloodeqtlbrowser/>), and BIOS QTL browser (<http://genenetwork.nl/biosqtlbrowser/>). Functionally-annotated SNPs were then mapped to genes based on physical position on the genome, eQTL associations (all tissues) and chromatin interaction mapping (all tissues). Intergenic SNPs were mapped to the two closest up- and down-stream genes which can result in their being assigned to multiple genes."

"15. When referring to loci that overlap between cognitive function and traits in the GWAS catalogue, how is overlap defined? Does it mean the lead variant from this study is in high r^2 with the lead SNP for the GWAS catalogue trait?"

Response: We looked up all independent significant SNPs and tagged SNPs (LD of $r^2 \geq 0.6$ with at least one of the independent significant SNPs) for general cognitive function that have previously been identified at $P < 9 \times 10^{-6}$ in GWAS of diseases and traits listed in the GWAS Catalog 2017-08-22.

Reviewer 3

“The paper describes GWAS analyses on cognitive function in over 280.000 participants. It uses an established proxy measure for general cognitive function that is based on two independent cognitive tests assessed in each cohort, thus enabling comparison of cohorts assessed with distinct cognitive measures. Using this approach it identifies 9714 SNPs of genome-wide significance, of which 120 were independent lead SNPs in 99 significant genomic loci. The authors thus have expertly generated an impressive dataset that is of immense potential value.

The paper then moves on to characterise these SNPs in a flurry of biometric analyses that may be informative, but do not provide much insight or develop hypotheses about the actual biological mechanisms that are identified by the genes discovered. This in my view is the main weakness of the paper.

For example, we learn through scores derived from CADD or Regulate DB that SNPs in the genetic loci might be deleterious or might affect function. However, there are no first hand data supporting and exactly quantifying these effects, nor is there a serious attempt (except for MAGMA gene set analyses providing an output at the most superficial level) to elucidate biochemical pathways and neural systems that may be affected by the functionality identified.

Thus the authors leave this extremely interesting dataset for others to interpret, and form and test hypotheses that can actually advance the field. This appears to be the explicit intention of the authors who describe their work as providing 'foundations for exploring mechanisms'. While there is merit in this approach there is also an element of dissatisfaction to this reviewer.

If the editors concur with the approach chosen by the authors I would not have any major comments about the expert execution of this work.”

Response: We are grateful to the referee for recognising the potential value of our findings. We agree that in the original manuscript we did not provide enough insight or develop hypotheses about the actual biological mechanisms that are identified by the genes discovered; this was in part because of the restrictions of the concise ‘Letter’ format, which is no longer necessary. We have now gone to considerable effort to highlight the biological mechanisms of interest and how they relate to existing knowledge. See p19-22.

Reviewer #1 (Remarks to the Author):

Overall, I think is a responsive revision. My main criticism of the original MS was that the descriptions of methodologies used in key analyses were sometimes not described with the level of detail I would have liked. I also wanted to know more about which findings are novel and which ones were not. The authors have done a lot of work in response to these criticisms.

Reviewer #2 (Remarks to the Author):

Overall question:

How do the results (loci, gene sets and conclusions) in this paper compare to the Hill et al. "A combined analysis of genetically correlated traits identifies 187 loci and a role for neurogenesis and myelination in intelligence", published online in January 2018 in Molecular Psychiatry. The gene sets identified are very similar. It seems likely the loci would be similar as well. The results in this paper need to be put in context of the Hill et al paper.

Specific comments:

1. The criteria to identify "independent" SNPs ($r^2 \leq .6$) does not identify independent SNPs. A strong signal can have more than one variant detected with $r^2 \leq .6$ and $p < 5 \times 10^{-8}$. The term independent SNPs can cause confusion because it would imply that there is more than one truly independent signal at many loci. An alternative way to start section would be with the number of lead SNPs found and the number of loci, and then provide the criteria for overlap with previous cognitive related results and with the other traits.
 2. Please remove the reference to FUMA in the results and give how SNPs were selected.
 3. Add to the results text the number of loci that were previously found for the studies corresponding to the confirmation numbers of 11, 24 and 89, ie 11 of xx, 24 of xx and 89 of xx were confirmed in this study. Add the Hill et al. (2018) paper to this section. Include in the discussion reasons why didn't these finding might not have found all previously reported signals.
 4. The gene-based test response is not convincing. 1) The Hill et al paper shows enrichment in coding regions. It is clear that coding regions contain SNPs that are causal for disease. Unlike coding variants, the presence of strongly associated SNPs in an intron does not imply the gene in which the SNP(s) is located is causal. 2) As the sample size increases, the signal in each locus will increase, leading to larger regions covered by strongly associated SNPs. This will lead to the "discovery" of more significantly associated genes in each locus, but not to more certainty about which (if any) associated gene is causal (using the current gene-based test over the body of a gene).
- A. Within the abstract, please remove the phrase "associated gene" when naming the genes of interest in newly identified loci.
- B. Please include a statement that gives perspective to the interpretation of the gene-based

association. It might be something like “association of a gene with cognitive function does not imply that it is causally related to the disease, only that the gene is in a region of strong association within a locus”.

5. Related to the point above, the revised text’s statement that “researchers could prioritize the genetic loci uncovered here that overlap with brain related measures” is ambiguous. Does it mean they could prioritize the associated genes or that they should use other means to prioritize variants for functional follow up?

6. Clarify in the methods for the gene-based tests that all SNPS within the gene-body, not just protein coding SNPs were used in the analysis.

7. The % variance explained by the first principal component ranges across studies from 24% to 65%. The lower numbers do not sound consistent with the presence of a single principal component. Were the principal component analysis of the COGENT cohorts checked in the same way as the CHARGE cohorts to establish the presence of a single principal component? What does “sufficient” mean in the statement “sufficient loading on the first unrotated principal component”?

8. How were the related individuals in the UK biobank study removed before analysis (~30% have relatedness of 3rd degree or greater)? How were they removed from the household-based Understanding Society study?

Reviewer comments

Reviewer 1

“Overall, I think is a responsive revision. My main criticism of the original MS was that the descriptions of methodologies used in key analyses were sometimes not described with the level of detail I would have liked. I also wanted to know more about which findings are novel and which ones were not. The authors have done a lot of work in response to these criticisms.”

Response: We thank the reviewer for this positive response to our revision.

Reviewer 2

“Overall question:

How do the results (loci, gene sets and conclusions) in this paper compare to the Hill et al. “A combined analysis of genetically correlated traits identifies 187 loci and a role for neurogenesis and myelination in intelligence”, published online in January 2018 in Molecular Psychiatry. The gene sets identified are very similar. It seems likely the loci would be similar as well. The results in this paper need to be put in context of the Hill et al paper. “

Response: We thank the reviewer for this comment. In the previous version of this paper we include all relevant comparisons to the Hill et al. paper, detailing all replicated findings (loci, gene-based, and gene-sets), in both the Results and Discussion sections. In the main text, the Hill et al. paper is Ref. 17. As suggested in specific comment 3, we have expanded the comparison of the 148 loci to include the number of loci that were found in previous studies.

We highlight these comparisons below:

Revised Text Pg 15: “A comparison of these 148 loci with results from the largest previous GWASs of cognitive function¹⁶, and educational attainment²⁴, and an MTAG analysis of cognitive function¹⁷— all of which included a subsample of individuals contributing to the present study—confirmed that 11 of 18, 24 of 74, and 89 of 187 of these were, respectively, genome-wide significant in the present study (Supplementary Table 15).”

Pg 16: “These 709 genes were compared to gene-based associations from previous studies of general cognitive function and educational attainment^{13,16,17,25}; 418 were replicated in the present study, and 291 were novel.”

Pg 20: “Upon additional exploration of the 58 newly-associated genetic loci, we find that many contain genes that are of further interest. All of the genes discussed below are also genome-wide significant in the general cognitive function gene-based association analysis ($P < 2.75 \times 10^{-6}$; Supplementary Table 7). Significant gene-based associations with general cognitive function have also been previously reported for GATAD2B, SLC39A1 and AUTS2^{16,17}”

Pg 20: “ATXN1L, ATXN2L and ATXN7L2 were also located in significant loci that have previously been associated with cognitive function, intelligence, or educational attainment^{16,17,24}.”

Pg 21: *Of the seven significant gene sets identified, one was a new finding: ‘positive regulation of nervous system development’. A more detailed description of this gene-set is: ‘any process that activates, maintains or increases the frequency, rate or extent of nervous system development, the origin and formation of nervous tissue’. The remaining six gene-sets showed replication with previous studies of general cognitive function and/or education^{16,17,24}. Only one, ‘regulation of cell development’, was significant across all four studies^{16,17,24}.*

“1. The criteria to identify “independent” SNPs ($r^2 \leq 0.6$) does not identify independent SNPs. A strong signal can have more than one variant detected with $r^2 \leq 0.6$ and $p < 5 \times 10^{-8}$. The term independent SNPs can cause confusion because it would imply that there is more than one truly independent signal at many loci. An alternative way to start section would be with the number of lead SNPs found and the number of loci, and then provide the criteria for overlap with previous cognitive related results and with the other traits. “

Response: We strongly agree with the reviewer that this use of “independent SNPs” is potentially confusing. However, in order to remain consistent with the terminology used by the package in which these analyses were performed, we have retained the term “independent”. To prevent possible confusion, we have provided a detailed description in the Methods section to clarify what is meant by “independent” within the context of this study. This description was in the previous version of this paper; however, we have now edited the following sentence in the Results section (Pg15) to direct the reader to this information:

Updated text Pg 15: *There were 434 ‘independent’ significant SNPs; see Methods for description of independent SNP selection criteria, distributed within 148 loci across all autosomal chromosomes. Note that, for consistency, we use the term ‘independent’ here according to the definition that is used in the relevant analysis package.*

“2. Please remove the reference to FUMA in the results and give how SNPs were selected.”

Response: We have removed the reference to FUMA in the results and replaced with “see Methods for description of independent SNP selection criteria”. The criteria for identifying independent, lead and tagged SNPs are described in full in the methods, we provide the existing text from the relevant section below.

Revised text Pg 15: *“There were 434 independent significant SNPs; see Methods for description of independent SNP selection criteria, distributed within 148 loci across all autosomal chromosomes.”*

Methods Text Pg 27: *“Genomic risk loci were defined from the SNP-based association results, using FUnctional Mapping and Annotation of genetic associations (FUMA)²³. Firstly, independent significant SNPs were identified using the SNP2GENE function and defined as SNPs with a P-value of $\leq 5 \times 10^{-8}$ and independent of other genome wide significant SNPs at $r^2 < 0.6$. Using these independent significant SNPs, tagged SNPs to be used in subsequent annotations were identified as all SNPs that had a MAF ≥ 0.0005 and were in LD of $r^2 \geq 0.6$ with at least one of the independent significant SNPs. These tagged SNPs included those from the 1000 genomes reference panel and need not have been included in the GWAS performed in the current study. Genomic risk loci that were 250kb or closer were merged into a single locus. Lead SNPs were also identified using the*

independent significant SNPs and were defined as those that were independent from each other at $r^2 < 0.1$.

“3. Add to the results text the number of loci that were previously found for the studies corresponding to the confirmation numbers of 11, 24 and 89, ie 11 of xx, 24 of xx and 89 of xx were confirmed in this study. Add the Hill et al. (2018) paper to this section. Include in the discussion reasons why didn't these finding might not have found all previously reported signals. “

Response: We have added the total number of loci previously reported by Sniekers *et al.*, Okbay *et al.*, and Hill *et al.* The Hill *et al.* 2018 paper (Ref. 17) was already included in this section in the previous version of the paper. We have now added text to the Discussion to address why we were unable to replicate all of the previously-reported findings.

Revised text Pg 15: ***“A comparison of these 148 loci with results from the largest previous GWASs of cognitive function¹⁶, and educational attainment²⁴, and an MTAG analysis of cognitive function¹⁷— all of which included a subsample of individuals contributing to the present study—confirmed that 11 of 18, 24 of 74, and 89 of 187 of these were, respectively, genome-wide significant in the present study (Supplementary Table 15).”***

Inserted text Pg23: ***“When compared to previous large studies of cognitive function and education, we replicate a large proportion, but not all, of the previously-reported significant findings. These differences in reported findings might be explained partly by differences in study populations (including age, social status, and ethnicity), phenotypes, and analysis methods. Whereas we know that there is sample overlap in the studies described, each comprises a unique set of contributing cohorts. As described above, there is substantial variation in the cognitive tests that contribute to the construction of a general cognitive function phenotype. Cognitive function is not as simple to measure as, say, height, and it is far from being standardised. This limitation applies across the GWAS meta-analysis studies, as well as within them. The use of different analysis methods—for example MTAG, which includes phenotypes other than the target phenotype—might also contribute to the different findings that have been reported. Finally, it is also possible that, although specific loci reached genome-wide significance in particular studies, there are false positives, highlighting the importance of well-powered replication studies. ”***

“4. The gene-based test response is not convincing. 1) The Hill et al paper shows enrichment in coding regions. It is clear that coding regions contain SNPs that are causal for disease. Unlike coding variants, the presence of strongly associated SNPs in an intron does not imply the gene in which the SNP(s) is located is causal. 2) As the sample size increases, the signal in each locus will increase, leading to larger regions covered by strongly associated SNPs. This will lead to the “discovery” of more significantly associated genes in each locus, but not to more certainty about which (if any) associated gene is causal (using the current gene-based test over the body of a gene).

A. Within the abstract, please remove the phrase “associated gene” when naming the genes of interest in newly identified loci.

B. Please include a statement that gives perspective to the interpretation of the gene-based association. It might be something like “association of a gene with cognitive function does not imply that it is causally related to the disease, only that the gene is in a region of strong association within a locus”.

Response A: We thank the reviewer for this comment and agree that “associated gene” is not an appropriate phrase to be used when describing genes located within loci identified by SNP-based association analyses. As suggested, we have removed the phrase “associated gene” from the abstract.

Revised text Pg 11: *“Genes within these loci included ATXN1, DCDC2, TTBK1, and CWF19L1.”*

Response B: We thank the reviewer for this comment and have added the suggested text.

Inserted text Pg 24: *“From these gene-based analyses, the association of a gene with general cognitive function does not imply that it is causally related to this phenotype, only that the gene is in a region of strong association within a locus. These loci may contain multiple associated genes; therefore, we note that all of the associated genes that we reported may not be independent findings.”*

“5. Related to the point above, the revised text’s statement that “researchers could prioritize the genetic loci uncovered here that overlap with brain related measures” is ambiguous. Does it mean they could prioritize the associated genes or that they should use other means to prioritize variants for functional follow up? “

Response: We thank the reviewer for highlighting the ambiguity in this statement. We have now expanded this to make our point clearer, and we have moved it to a more appropriate place within the discussion.

Revised text Pg 22:*“Researchers following up the present study’s results could prioritize the genetic loci uncovered herein that are associated with general cognitive function and reaction time (Supplementary Tables 9 and 10), as well as those that are also associated with brain-related measures in other large GWASs. Such variants, being associated with multiple cognitive and neurological phenotypes, might help to prioritise potentially causal variants, and help to identify how differences in genotypic sequence are linked to such phenotypic consequences.”*

“6. Clarify in the methods for the gene-based tests that all SNPs within the gene-body, not just protein coding SNPs were used in the analysis.”

Response: We have clarified this in the methods section for gene-based tests.

Revised text Pg 28: *“SNPs were mapped to genes based on genomic location. All SNPs that were located within the gene-body were used to derive a P-value describing the association found with general cognitive function and reaction time.”*

“7. The % variance explained by the first principal component ranges across studies from 24% to 65%. The lower numbers do not sound consistent with the presence of a single principal component. Were the principal component analysis of the COGENT cohorts checked in the same way as the CHARGE cohorts to establish the presence of a single principal component? What does “sufficient” mean in the statement “sufficient loading on the first unrotated principal component”?”

Response: Those cohorts with a relatively low percentage of variance accounted for by the first unrotated principal component were compatible with a general cognitive component. The reason is that they had relatively large numbers of cognitive tests. For example, consider those cohorts where the percentage of variance accounted for was below 30% (there were only two, both from COGENT). The Duke Neurogenetics Study (24.2% variance

accounted for) had 12 tests, and the NIMH Genes, Cognition and Psychosis Program (27.3% variance accounted for) had 14 tests. In the first case one can calculate that the eigenvalue was about 2.9, and in the second it is about 3.8, which does indicate a first unrotated principal component accounting for substantial general cognitive variation. Sufficient loadings were at least 0.3, but almost all individual tests' loadings were well above that (though, of course, it is well known that some tests have lower loadings on the general cognitive function component than others). The COGENT method was also principal components analysis, and this is described in Trampush *et al.* (2017, *Molecular Psychiatry*, 22, 336-345, pp. 337-338, and Supplementary Table 1). This has now been added to the Participants and Cognitive Phenotypes section of the Methods, as follows.

Inserted text Pg 25: "Principal component analyses for the COGENT cohorts are described in Trampush et al. (pp. 337-338, and Supplementary Table 1).⁶⁴"

"8. How were the related individuals in the UK biobank study removed before analysis (~30% have relatedness of 3rd degree or greater)? How were they removed from the household-based Understanding Society study? "

Response: For UK Biobank, all related individuals were identified using the UK Biobank QC variable for relatedness ("in.kinship.table" as described in Bycroft *et al.* 2017). To maximise our sample size of unrelated individuals, we created a GRM of those related individuals (N = 131,790) and then removed one of a pair of individuals based on a genetic relationship threshold of 0.025 ascertained using GCTA-GREML. GCTA selectively removes individuals to maximize the remaining sample size. In the UK Biobank cohort description (Supplementary Material Section 1) we now describe these QC procedures in more detail.

Inserted text Pg 45 of Supplementary Information: "Full details of the UK Biobank genotyping procedure can be found elsewhere (Bycroft et al. 2017). In short, two custom genotyping arrays were used to genotype 49,950 participants (UK BiLEVE Axiom Array) and 438,427 participants (UK Biobank Axiom Array) (Bycroft et al. 2017, Wain et al. 2017). Genotype data (805,426 markers) were available for 488,377 individuals, with imputation to the HRC reference panel¹¹. Downstream quality control steps for the present study involved excluding (1) those with non-British ancestry based on self-report and a principal components analysis, (2) extreme scores based on heterozygosity and missingness, (3) individuals with neither XX nor XY sex chromosomes, (4) individuals whose reported sex was inconsistent with genetically inferred sex, and (5) individuals with >10 putative third degree relatives from the kinship table. This left 408,095 individuals. All related individuals were identified using the UK Biobank QC variable for relatedness ("in.kinship.table" as described in Bycroft et al. 2017). To maximise our sample size of unrelated individuals, we created a GRM of those related individuals (N = 131,790) and then removed one of a pair of individuals based on a genetic relationship threshold of 0.025 ascertained using GCTA-GREML (Yang et al. 2011). After implementing these steps, the sample size was 332,050, with verbal numerical reasoning data available for 168,033 individuals."

For Understanding Society, we were provided with a variable ("GRMindp") to facilitate the removal of individuals who are more than 5% related. This resulted in 707 individuals being removed from the Understanding Society dataset prior to any analyses. We have now added this to the Understanding Society cohort description (Supplementary Material Section 1).

Inserted text Pg 47: "We were provided with a variable ("GRMindp"), which allows identification of related individuals within the study. This was used to remove those who are more than 5% related (n = 707)."

Reviewer #2 (Remarks to the Author):

The authors have been responsive to almost all the comments and have improved the paper. I have no further comments.